# DON'T NEED RETRAINING:
# A Mixture of DETR and Vision Foundation Models for Cross-Domain Few-Shot Object Detection

**Changhan Liu, Xunzhi Xiang, Zixuan Duan, Wenbin Li, Qi Fan$^{\boxtimes}$, Yang Gao**

School of Intelligence Science and Technology, Nanjing University, China
https://github.com/lch216/VFMDETR

## Abstract

Cross-Domain Few-Shot Object Detection (CD-FSOD) aims to generalize to unseen domains by leveraging a few annotated samples of the target domain, requiring models to exhibit both strong generalization and localization capabilities. However, existing well-trained detectors typically have strong localization capabilities but suffer from limited generalization, whereas vision foundation models (VFMs) generally exhibit better generalization but lack accurate localization capabilities. In this paper, we propose a novel Mixture-of-Experts (MoE) structure that integrates the detector's localization capability and the VFM's generalization by using VFM features to improve detector features. Specifically, we propose Expert-wise Router (ER) that dynamically selects the most relevant VFM experts for each backbone layer, and Region-wise Router (RR) that emphasizes foreground and suppress background. To bridge representation gaps, we further propose Shared Expert Projection (SEP) module and Private Expert Projection (PEP) module, which align VFM features to the detector feature space while decoupling shared image feature from private image feature in the VFM feature map. Finally, we construct MoE module to transfer the VFM's generalization to the detector without modifying the original detector architecture. Furthermore, our method extend well-trained detectors for detecting novel classes in unseen domains without re-training on the base classes. Experimental results on multiple cross-domain datasets validate the effectiveness of our method.

## 1 Introduction

Few-Shot Object Detection (FSOD) aims to detect objects of novel classes with a few labeled support data. Existing FSOD methods [1, 2, 3, 4, 5, 6, 7, 8] have achieved notable progress in generalizing to various in-domain novel categories. However, these methods often struggle with domain shifts, where training and testing data originate from different domains. This challenge underscores the significance of Cross-Domain Few-Shot Object Detection (CD-FSOD). CD-FSOD aims to generalize object detection models to detect novel classes in unseen domains by using a few training samples. This challenging task typically requires model to combine strong generalization and accurate localization capability.

Existing well-trained object detection methods [9, 10, 11, 12, 13, 14, 15] usually excel at localizing and recognizing seen objects but struggle to generalize to unseen categories or knowledge. To address this problem, previous CD-FSOD methods [16, 17, 18] mainly adopt a two-stage training paradigm that first involves base training on a large-scale dataset of base classes, follow by novel fine-tuning

---

$^{\boxtimes}$ indicates corresponding author.

39th Conference on Neural Information Processing Systems (NeurIPS 2025).

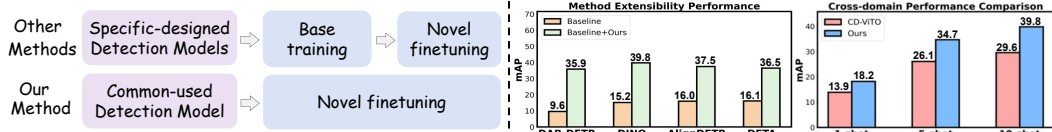

Figure 1: Existing CD-FSOD methods generally require time-consuming base training to adapt their specific-designed detection models. In contrast, our method directly extends the well-trained common-used detection models, *e.g.*, DETR [25], to CD-FSOD without retraining the model on base data. Our method significantly improves the cross-domain performance of well-trained object detection methods [21, 20, 19, 22]. Our method also outperforms the state-of-the-art method CD-ViTO [18].

on a small dataset of novel classes. However, these models are typically redesigned for CD-FSOD and retrained on base classes which brings substantial computational and time costs, as shown in Figure 1. Moreover, they fail to fully utilize the well-trained powerful object detection models, *e.g.*, DETR-based methods [19, 20, 21, 22]. Thus, we propose a novel CD-FSOD paradigm by extending well-trained in-domain object detection methods to detect unseen classes in new domains without retraining on the base classes. Our key idea is to leverage the powerful vision foundation models (VFMs) [23, 24] to equip well-trained DETR models with powerful domain generalization ability.

Vision foundation models [26, 27] have demonstrated excellent generalization to new domains, powered by sophisticated architectures and large-scale pretraining datasets. However, VFMs typically lack accurate localization capabilities due to the lack of bounding box annotations in their pretraining datasets. Consequently, directly applying VFMs to object detection generally results in suboptimal performance. To address this problem, existing VFM-based CD-FSOD methods [18, 28] usually modify the detector structure to integrate VFMs and retrain on the base data to achieve precise localization capabilities, which incurs substantial computational and time costs.

It is essential to fully utilize the VFM's generalization capabilities and the well-trained object detection model's localization capabilities while reducing computational and time costs. Thus, instead of altering the structure of existing detection models, we introduce a flexible framework based on a Mixture-of-Experts (MoE) [29] architecture to integrate detector with VFM. Our method enables well-trained detectors to achieve both strong localization and generalization capabilities, without retraining on base class. Specifically, we adopt Detection Transformer (DETR) as our detection framework due to its superior localization capabilities and leverage VFM features as experts to improve the detector's representations by aggregating both detector and VFM features.

Overall, our method focuses on two parts: *VFM expert feature selection* and *VFM–detector feature fusion*. Since detector features at different layers contain varying levels of semantic information, each backbone layer generally requires distinct VFM features for effective guidance. Accordingly, we propose *Expert-wise Router (ER)* to select appropriate VFM features based on image features from different backbone layers. Meanwhile, to highlight the foreground regions and suppress irrelevant background in the VFM feature map, we propose the *Region-wise Router (RR)* module which generates region-wise gating weights to reweight different spatial regions in the VFM expert features. The ER and RR modules respectively generate expert-wise and region-wise gating weights, dynamically prioritizing the most relevant VFM expert features for detector features at different layers and effectively suppressing irrelevant background in the VFM expert features.

After selecting expert features, it is essential to ensure effective feature fusion between VFM expert features and detector features. Due to the differences in both channel and spatial dimensions between VFM expert features and detector features, it is necessary to project VFM features into the detector feature space. However, assigning a dedicated projection layer to each expert feature will introduce a large number of additional parameters, which increases the computational cost. In contrast, using a single shared projection layer fails to capture the diverse projection requirements of different expert features. To address this issue, we propose the *Shared Expert Projection (SEP)* and *Private Expert Projection (PEP)* modules to project the VFM features to the detector feature space with minimal parameters. Each expert feature contains shared image features, such as object shape. Additionally, since each expert feature may focus on different regions of the object, it also includes private image features. Based on this characteristic, we decouple the shared and private features of each expert feature and project them using separate modules. Specifically, we use the SEP module to project

shared features across all expert features, while using the PEP module to project the private features that are unique to each expert feature. This design minimizes the number of parameters required for feature projection and preserves detailed features from different object regions.

Compared with existing methods that adopt the foundation model as a learnable backbone [30, 31, 32] or a frozen backbone [33, 34], our method offers the following advantages:

- **Maintaining strong localization capability.** Our method uses VFM features as experts to enhance detector backbone features, rather than using the VFM as the detector backbone directly. This design avoids transferring the VFM's limited localization capability to the detector.
- **No retraining requirement on base classes.** We introduce only a few trainable parameters to fuse VFM and detector features. Therefore, the model can be directly finetuned on downstream tasks without retraining on base classes.
- **High extensibility.** By treating the VFM as an expert model independent of the detector, our method avoids modifying the detector structure and can be easily transferred to other well-trained object detection models.

## 2 Related Works

**DETR and Its Variants.** Detection Transformer (DETR)[25] initially proposed an end-to-end object detector based on the Vision Transformer architecture. Many subsequent studies [35, 36] have improved DETR from various perspectives, such as accelerating convergence speed [37, 38, 20, 39], improving the matching strategy [38, 40], and applying advanced query learning methods [41, 21, 42, 43]. Although these methods demonstrate strong performance, DETR and its variants typically struggle on CD-FSOD tasks because they are trained on base classes and thus fail to extract robust representations for target domain images, impairing DETR's generalization capabilities. In this work, we propose a MoE framework to integrate DETR with vision foundation model. Specifically, we leverage vision foundation model's features as experts to improve the detector's features, enabling the detector to inherit the strong generalization of vision foundation model.

**Cross-Domain Few-Shot Object Detection.** Cross-Domain Few-Shot Object Detection (CD-FSOD) aims to enable object detection models to detect novel classes in unseen domains by using only a few training samples. Existing FSOD methods can be categorized into meta-learning-based [44, 34, 45], fine-tuning-based [46, 47, 48, 49], and data-enhancement-based [50, 51, 52, 53] approaches. Although these methods perform well in standard FSOD tasks, their performance generally degrades in CD-FSOD settings due to substantial domain gaps. Several methods [54, 55] specifically address CD-FSOD challenges. For instance, AcroFOD [16] uses domain-aware augmentation to reduce domain gaps. OA-FSUI2IT [17] leverages a few unlabeled target-domain samples to generate cross-domain images for feature alignment. CD-ViTO [18] improves inter-class discrimination using DINOv2 prototypes. ETS [56] uses a two-step augmentation strategy for foundation model adaptation. CDFormer [57] introduces transformer-based modules to distinguish object-background and object-object features. IFC [58] uses learnable instance feature caches for robust prototypes. These methods generally require modifying the original detector's structure and retraining on the base classes. In this work, we propose a novel Mixture-of-Experts (MoE) structure to aggregate VFM features and detector features with a few parameters. Our method treats the VFM as an expert model independent of the detector, avoiding modifications to the detector architecture and reducing the computational and time cost of retraining.

**Vision Foundation Model.** Current vision foundation models (VFMs) are primarily trained under two paradigms: supervised learning and self-supervised learning. Supervised learning model [59, 60] commonly relies on high-quality labeled datasets for pretraining. These models often exhibit strong generalization, demonstrating excellent performance even without fine-tuning based on downstream tasks. However, large-scale, high-quality labeled datasets are difficult to obtain due to the high cost of manual annotation. To address this problem, self-supervised learning models [61, 62, 63, 64, 65, 66] leverage techniques such as contrastive learning [67, 68] and masked image modeling [69, 70] to train on unlabeled data, reducing reliance on labeled datasets and improving image understanding through greater data diversity. However, the lack of accurate bounding boxes limits the localization capabilities of vision foundation models. When the vision foundation model is directly used as the backbone for object detection tasks, its weak localization capability is transferred to the detector.

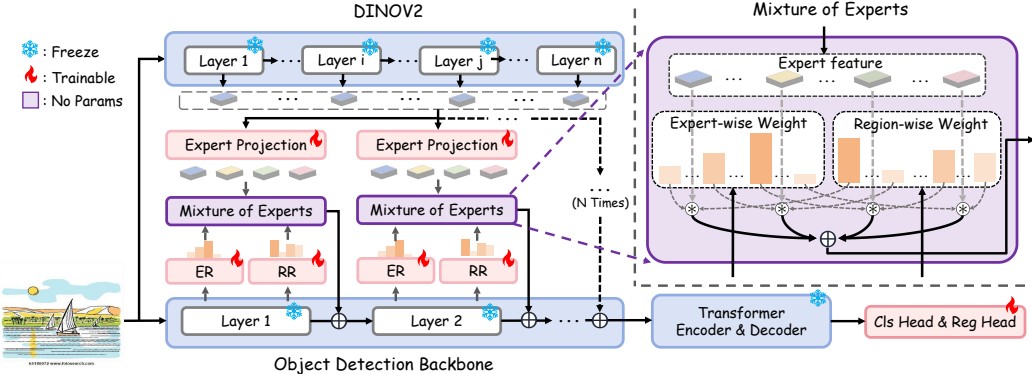

Figure 2: Method overview. Our method consists of two main components: 1) : The Expert-wise Router (ER) module generates expert-wise gating weight to select the appropriate VFM expert features for the detector's features at different layers. The Region-wise Router (RR) module generates region-wise gating weight to filter out the invalid background regions in the VFM feature map. 2) : Expert Projection (EP) module: The EP module primarily consists of shared expert projection (SEP) module and private expert projection (PEP) module. The SEP module projects the shared image feature contained in different expert features. The PEP module projects the private image feature contained in each expert feature.

To overcome this limitation, we use VFM features as experts to improve the detector backbone features, rather than directly replacing the detector's backbone. This approach avoids modifications to the detector architecture and preserves the detector's localization capability. Additionally, since our method does not modify the detector structure, it can be easily applied to other detector architectures.

## 3 Method

### 3.1 Method Overview

To fully exploit the VFM's generalization for CD-FSOD while preserving the detector's strong localization capability, we propose a novel MoE framework that leverages VFM features to improve detector representations without modifying the original detector architecture, as shown in Figure 2. It consists of two key components: the expert routing module and the expert projection modules. Specifically, given VFM expert features $\{\boldsymbol{F}_d^n\}_{n=1}^N \in \mathbb{R}^{B \times C_d \times H_d \times W_d}$ and detector feature $\mathbf{F}^l \in \mathbb{R}^{B \times C \times H \times W}$ from the $l$-th detector backbone layer, where $B$ denotes the batch size, $N$ corresponds to the number of expert features, $C, C_d$ represent the channel dimension of detector feature and VFM expert feature, $H \times W, H_d \times W_d$ represent the spatial size of detector feature and VFM expert feature, the Expert-wise Router (ER) module is used to generate expert-wise gating weights base on $\mathbf{F}^l$ to select suitable VFM expert features for current detector feature. Simultaneously, the Region-wise Router (RR) module is used to generate region-wise gating weights based on $\mathbf{F}^l$ to suppress irrelevant background regions in the selected VFM expert feature. Then, the Shared Expert Projection (SEP) and Private Expert Projection (PEP) modules are applied to address the dimensional mismatch between the VFM features and the detector features. The SEP module is used to project the shared feature contained in expert features. The PEP module is used to project the private feature contained in each expert feature. After applying expert projection, the projected VFM features are denoted as $\{\boldsymbol{F}_{d'}^{n,l}\}_{n=1}^N \in \mathbb{R}^{B \times C \times H_d \times W_d}$. Finally, the MoE module fuses VFM features and detector features to obtain $\boldsymbol{F}_f^l \in \mathbb{R}^{B \times C \times H \times W}$ as the input for the next layer of the detector backbone.

### 3.2 Expert Routers

**Expert-wise Router.** Detector features at different layers contain different levels of image features. Shallow layers primarily focus on low-level feature such as edges and color, and deep layers focus on high-level semantics such as object shape and category. Therefore, the importance of each VFM expert feature varies across different layers of the detector backbone. Inspired by the routing strategy of MoE, we propose the expert-wise router to select the most appropriate VFM expert features for detector features at different backbone layers. Specifically, for the $l$-th layer detector feature $\mathbf{F}^l$, we first apply

global average pooling along the spatial dimensions to aggregate spatial information of detector feature and obtain spatial-wise aggregated feature vector $\mathbf{f}_g = \frac{1}{H \times W} \sum_{i=1}^{H} \sum_{j=1}^{W} \mathbf{F}^l(:,:,i,j)$. Then, the tensor $\mathbf{f}_g$ is fed to a learnable fully connected layer $\boldsymbol{\theta}_e$ to aggregate channel information of detector feature and obtain channel-wise aggregated feature vector $\mathbf{f}'_g = \boldsymbol{\theta}_e(\mathbf{f}_g)$. Finally, a softmax function is applied to the channel-wise aggregated feature vector $\mathbf{f}'_g$ along channel dimension to obtain the expert-wise gating weight $\mathbf{G}_e = \mathrm{softmax}\left(\mathbf{f}'_g\right)$.

The expert-wise router generates expert-wise gating weights to select the most suitable VFM expert features for different backbone layers. These VFM expert features typically exhibit high similarity to the detector features, indicating they are more likely to focus on similar levels of image semantics. Therefore, the expert-wise router can effectively improve the detector's feature of different layers.

**Region-wise Router.** Different regions in the VFM feature maps exhibit varying importance for the detector. Foreground regions are typically more informative for object detection and background regions often contain more task-irrelevant image features. To filter out irrelevant background regions and highlight foreground objects, we propose the region-wise router to generate region-wise gating weights for different regions of the VFM expert feature maps. Specifically, the $l$-th layer detector feature $\mathbf{F}^l$ is fed to a learnable fully connected layer $\boldsymbol{\theta}_r$ along channel dimension to obtain projected tensor $\mathbf{f}_r = \boldsymbol{\theta}_r(\mathbf{F}^l)$. The projected tensor $\mathbf{f}_r$ is then normalized using a softmax function to generate the region-wise gating weights $\mathbf{G}_r = \mathrm{softmax}\left(\mathbf{f}_r\right)$.

The region-wise router generates region-wise gating weights to reweight different regions of the VFM feature map. Compared with VFM, the detector can precisely localize foreground regions because of its stronger localization capability. Therefore, The region-wise router generates region-wise gating weights based on the detector features to highlight foreground regions.

## 3.3 Expert Projections

Since VFM expert features and detector features differ in both spatial and channel dimensions, it is necessary to project VFM expert into the detector feature space to align their feature dimensionality. However, due to the large number of VFM experts, assigning a dedicated projection layer to each VFM expert feature would introduce substantial parameters and degrade training efficiency. In contrast, using a single shared projection layer reduces parameter overhead but fails to accommodate the diverse projection needs across VFM expert features. To address this problem, we propose the Shared Expert Projection (SEP) module and Private Expert Projection (PEP) module. Each expert feature contains both shared information, *e.g.*, object shape, and private information, *e.g.*, fine-grained details from different object regions. The SEP module is used to project shared information across all VFM expert features, and the PEP module is used to project private information specific to each VFM expert feature.

**Shared Expert Projection.** The shared expert projection module enables the model to effectively project shared image feature across different expert features. Specifically, for the $i$-th VFM expert feature $\boldsymbol{F}_d^i$, we apply a shared linear projection layer $\boldsymbol{\theta}_s \in \mathbb{R}^{C_d \times C_s}$ to all expert features, where $C_s = \frac{m-1}{m} C$, $m$ is hyperparameter that control the channel dimensions of the shared expert projection module transformation. This produces the shared projection component $\boldsymbol{F}_s^i = \boldsymbol{F}_d^i \cdot \boldsymbol{\theta}_s$.

**Private Expert Projection.** The private expert projection modules enable the model to project private image features contained in each VFM expert feature. Specifically, each expert feature applies its own private projection layer $\boldsymbol{\theta}_p^i \in \mathbb{R}^{C_d \times C_p}$ to project its private image features, where $C_p = \frac{1}{m} C$. The private projection component $\boldsymbol{F}_p^i$ can be obtained as $\boldsymbol{F}_p^i = \boldsymbol{F}_d^i \cdot \boldsymbol{\theta}_p^i$.

The projected VFM expert feature $\boldsymbol{F}_{d'}^i$ is constructed by concatenating $\boldsymbol{F}_s^i$ and $\boldsymbol{F}_p^i$ along the channel dimension. After projecting the VFM expert features along the channel dimension, we resize their spatial size via bilinear interpolation to match the spatial resolution of $l$-th detector backbone feature. The final VFM expert features at $l$-th detector backbone layer are denoted as $\{\boldsymbol{F}_{d'}^{n,l}\}_{n=1}^{N}$.

## 3.4 Mixture of Experts

Given the projected expert features $\{\boldsymbol{F}_{d'}^{n,l}\}_{n=1}^{N}$ and their corresponding expert-wise and region-wise gating weights for the $l$-th detector backbone layer, the final feature fusion is performed by our MoE

Table 1: Comparison results on the CD-FSOD benchmark. The models are trained on COCO dataset and evaluated on six datasets with distinct domain shifts. The best results are highlighted with **bold**. † indicates that the methods are fine-tuned on six cross domain datasets.

| | Methods | Backbone | ArTaxOr | Clipart1k | DIOR | DeepFish | NEU-DET | UODD | Average |
|---|---|---|---|---|---|---|---|---|---|
| **1-shot** | Distill-cdfsod† [54] | ResNet50 | 5.1 | 7.6 | 10.5 | - | - | 5.9 | - |
| | DINO DETR† [20] | ResNet50 | 2.9 | 13.6 | 6.9 | 11.6 | 4.5 | 2.8 | 7.1 |
| | ViTDeT† [72] | ViT-B/14 | 5.9 | 6.1 | 12.9 | 0.9 | 2.4 | 4.0 | 5.4 |
| | Detic [73] | ViT-L/14 | 0.6 | 11.4 | 0.1 | 0.9 | 0.0 | 0.0 | 2.2 |
| | Detic† [73] | ViT-L/14 | 3.2 | 15.1 | 4.1 | 9.0 | 3.8 | 4.2 | 6.6 |
| | DE-ViT [28] | ViT-L/14 | 0.4 | 0.5 | 2.7 | 0.4 | 0.4 | 1.5 | 1.0 |
| | DE-ViT† [28] | ViT-L/14 | 10.5 | 13.0 | 14.7 | 19.3 | 0.6 | 2.4 | 10.1 |
| | CD-ViTO† [18] | ViT-L/14 | 21.0 | 17.7 | 17.8 | 20.3 | 3.6 | 3.1 | 13.9 |
| | **Ours†** | ResNet50 | **26.1** | **20.1** | **20.6** | **24.2** | **9.1** | **9.0** | **18.2** |
| **5-shot** | Distill-cdfsod† [54] | ResNet50 | 12.5 | 23.3 | 19.1 | 15.5 | 16.0 | 12.2 | 16.4 |
| | DINO DETR† [20] | ResNet50 | 8.5 | 21.2 | 12.3 | 16.2 | 9.6 | 8.7 | 12.8 |
| | ViTDeT† [72] | ViT-B/14 | 20.9 | 23.3 | 23.3 | 9.0 | 13.5 | 11.1 | 16.9 |
| | Detic [73] | ViT-L/14 | 0.6 | 11.4 | 0.1 | 0.9 | 0.0 | 0.0 | 2.2 |
| | Detic† [73] | ViT-L/14 | 8.7 | 20.2 | 12.1 | 14.3 | 14.1 | 10.4 | 13.3 |
| | DE-ViT [28] | ViT-L/14 | 10.1 | 5.5 | 7.8 | 2.5 | 1.5 | 3.1 | 5.1 |
| | DE-ViT† [28] | ViT-L/14 | 38.0 | 38.1 | 23.4 | 21.2 | 7.8 | 5.0 | 22.3 |
| | CD-ViTO† [18] | ViT-L/14 | 47.9 | 41.1 | 26.9 | 22.3 | 11.4 | 6.8 | 26.1 |
| | **Ours†** | ResNet50 | **63.3** | **45.1** | **32.1** | **29.5** | **19.0** | **19.6** | **34.7** |
| **10-shot** | Distill-cdfsod† [54] | ResNet50 | 18.1 | 27.3 | 26.5 | 15.5 | 21.1 | 14.5 | 20.5 |
| | DINO DETR† [20] | ResNet50 | 11.4 | 23.2 | 14.4 | 20.5 | 11.8 | 9.9 | 15.2 |
| | ViTDeT† [72] | ViT-B/14 | 23.4 | 25.6 | 29.4 | 6.5 | 15.8 | 15.6 | 19.4 |
| | Detic [73] | ViT-L/14 | 0.6 | 11.4 | 0.1 | 0.9 | 0.0 | 0.0 | 2.2 |
| | Detic† [73] | ViT-L/14 | 12.0 | 22.3 | 15.4 | 17.9 | 16.8 | 14.4 | 16.5 |
| | DE-ViT [28] | ViT-L/14 | 9.2 | 11.0 | 8.4 | 2.1 | 1.8 | 3.1 | 5.9 |
| | DE-ViT† [28] | ViT-L/14 | 49.2 | 40.8 | 25.6 | 21.3 | 8.8 | 5.4 | 25.2 |
| | CD-ViTO† [18] | ViT-L/14 | 60.5 | 44.3 | 30.8 | 22.3 | 12.8 | 7.0 | 29.6 |
| | **Ours†** | ResNet50 | **71.3** | **49.9** | **37.8** | **34.1** | **23.7** | **22.1** | **39.8** |

module:

$$\boldsymbol{F}_{\mathrm{f}}^{l} = \boldsymbol{F}^{l} + \sum_{n=1}^{N} \left( \alpha \cdot \boldsymbol{G}_{\mathrm{e}}^{n,l} \circledast \boldsymbol{F}_{d'}^{n,l} + \beta \cdot \boldsymbol{G}_{\mathrm{r}}^{n,l} \circledast \boldsymbol{F}_{d'}^{n,l} \right), \tag{1}$$

where $\circledast$ denotes tensor broadcast multiplication, and $\alpha$ and $\beta$ are weighting factors, set to $\alpha = 0.5$, $\beta = 0.5$ in our experiment. $\boldsymbol{F}_{\mathrm{f}}^{l}$ denotes the output of the $l$-th detector's backbone layer. Simultaneously, it also serves as the input to layer $l + 1$. We repeat this process across all backbone layers until obtaining the final layer backbone features, which are then fed into the subsequent detector components to obtain the object detection results.

## 4  Experiments

We adopt the DINO detector [20] which is trained on the COCO source domain dataset as our baseline and use the self-supervised foundation model DINOv2 [27] as our expert model. Our method is directly applied to the publicly available DINO with a ResNet50 [71] backbone, without any additional re-training on the source domain dataset. For fine-tuning, we employ the AdamW optimizer with a learning rate of 2e-3. Following the benchmark in previous work [18], we evaluate our method on six datasets with distinct domain shifts: ArTaxOr (insect images), Clipart1k (hand-drawn cartoon image), DIOR (remote sensing imagery), DeepFish (underwater fish images), NEU-DET (industrial defect images), and UODD (marine organism images). For the 1/5/10 shot task settings, we train our model for 400, 800, and 1200 iterations, respectively. All experiments are conducted on four NVIDIA RTX 4090 GPUs. We rename the detector DINO as DINO DETR in the following.

### 4.1  Quantitative Results

**Comparison with State-of-The-Arts.** As shown in Table 1, we compare our method with typical CD-FSOD [18, 54], ViT-based OD [72], and open-set based OD/FSOD methods [73, 28]. Our method outperforms the baseline method DINO DETR, achieving improvements of 11.1/21.9/24.6 mAP on six cross domain datasets under the 1/5/10 shot task settings, respectively. Additionally, compared with the previous state-of-the-art cross-domain few-shot object detection method CD-ViTO,

Table 2: Comparison results of our method, MLLMs and OVMs under 10-shot task setting. The best results are highlighted in **bold**.

| Methods | ArTaxOr | Clipart1k | DIOR | DeepFish | NEU-DET | UODD | Average |
|---|---|---|---|---|---|---|---|
| Qwen model [74] | 48.8 | 7.5 | 2.7 | 9.2 | 4.5 | 1.3 | 12.3 |
| Ferret model [75] | 5.5 | 8.5 | 0.8 | 5.0 | 0.6 | 1.4 | 3.6 |
| YOLO-World [77] | 10.5 | 37.5 | 3.1 | 29.5 | 0.1 | 0.2 | 13.5 |
| Grounding DINO (Swin-B) [76] | 12.8 | 49.1 | 4.5 | 28.6 | 1.2 | 10.1 | 17.7 |
| DINO DETR (ResNet50) + Ours | **71.3** | **49.9** | **37.8** | **34.1** | **23.7** | **22.1** | **39.8** |

Table 3: Result of method extensibility. All models are trained on COCO. The best results on each baseline are highlighted in **bold**.

| Methods | Backbone | ArTaxOr | Clipart1k | DIOR | DeepFish | NEU-DET | UODD | Average |
|---|---|---|---|---|---|---|---|---|
| DAB-DETR [21] | ResNet50 | 8.2 | 19.4 | 8.2 | 9.7 | 6.9 | 6.1 | 9.6 |
| DAB-DETR + Ours | ResNet50 | **68.7** | **45.2** | **31.8** | **27.5** | **20.1** | **22.1** | **35.9** |
| DETA [22] | ResNet50 | 12.2 | 23.4 | 15.0 | 20.0 | 11.6 | 14.1 | 16.1 |
| DETA + Ours | ResNet50 | **69.9** | **45.5** | **37.1** | **26.3** | **20.9** | **19.0** | **36.5** |
| AlignDETR [19] | ResNet50 | 12.1 | 23.7 | 16.1 | 20.8 | 12.3 | 10.7 | 16.0 |
| AlignDETR + Ours | ResNet50 | **72.1** | **45.6** | **35.5** | **27.7** | **21.7** | **22.1** | **37.5** |

Table 4: Comparison results of different backbone. All models are trained on COCO. The best results are highlighted in **bold**.

| Methods | Backbone | ArTaxOr | Clipart1k | DIOR | DeepFish | NEU-DET | UODD | Average |
|---|---|---|---|---|---|---|---|---|
| DINO DETR + Ours | ResNet50 | 71.3 | 49.9 | 37.8 | 34.1 | 23.7 | 22.1 | 39.8 |
| DINO DETR + Ours | Swin-B | 75.4 | 56.7 | 39.5 | 35.1 | 23.2 | 23.1 | 42.2 |
| DINO DETR + Ours | ViT-L/14 | **75.8** | **60.3** | **42.0** | **37.2** | **25.1** | **25.9** | **44.4** |

our method achieves the improvements of 4.3/8.6/10.2 mAP on six cross domain datasets under the 1/5/10 shot task settings.

**Comparison with MLLMs and OVMs.** As shown in Table 2, we compare our method with multimodal large language models (MLLMs) and open-vocabulary methods (OVMs). Using their open-source code, we conducted fair comparisons on the same dataset. Qwen model [74] and Ferret model [75] obtain results through text-guided visual grounding. Grounding DINO [76] and YOLO-World [77] derive detection results via image-text matching. Our method achieves the highest performance compared with OVMs and MLLMs across six cross domain datasets. Compared with Grounding-DINO and YOLO-World model, our method achieves the improvements of 22.1/26.3 mAP. For Qwen model and Ferret model, our method achieves the improvements of 27.5/36.2 mAP. We argue that models such as Qwen, despite being trained on large-scale image-text datasets, have never seen the novel classes in the cross domain dataset, resulting in weak zero-shot performance. In contrast, our method adopts a cross-domain learning strategy that integrates with vision foundation models, achieving superior performance on cross-domain tasks.

**Method Extensibility Analysis.** To further validate the strong extensibility of our method, we adapt our method to other DETR models and evaluate their performance under the 10-shot task setting on all datasets. As shown in Table 3, the average performance of all models has been significantly improved. For instance, the performance of DAB-DETR increased from 9.6 mAP to 35.9 mAP. To validate the effectiveness of our method across different detector backbones, we apply our approach to DINO DETR with various backbones and evaluate their performance under the 10-shot task setting on six cross domain datasets. As shown in Table 4, the experimental results demonstrate that our method achieves strong performance across different backbones. Specifically, when integrating our method with a Swin-B [78] backbone, our method performance improves by 2.4 mAP compared to the ResNet backbone, while using a ViT-L [79] backbone leads to an improvement of 4.6 mAP.

**Ablation Studies.** To assess the contribution of each module, we conduct ablation studies under the 10-shot task setting across six cross domain datasets. As shown in Table 5, adding the SEP module raises the average performance to 34.6 mAP, surpassing the state-of-the-art CD-ViTO. Building on this, adding the PEP module further improves performance to 35.7 mAP, demonstrating that the private projection layer effectively retains rich private image features in the VFM expert features. After introducing the feature projection modules, we further evaluate the performance of the routing modules. Adding the expert-wise routing (ER) module improves the average performance to 37.2 mAP, highlighting the importance of selecting different VFM expert features to guide different detector layers. Adding the region-wise routing (RR) module further boosts performance to 38.2

Table 5: Results of ablation studies. "SEP" denotes the shared expert projection, "PEP" denotes the private expert projection, "ER" denotes the expert-wise router, "RR" denotes the region-wise router.

| SEP | PEP | ER | RR | ArTaxOr | Clipart1k | DIOR | DeepFish | NEU-DET | UODD | Average |
|-----|-----|-----|-----|---------|-----------|------|----------|---------|------|---------|
| × | × | × | × | 11.4 | 23.2 | 14.4 | 20.5 | 11.8 | 9.9 | 15.2 |
| ✓ | × | × | × | 62.1 | 43.8 | 34.5 | 26.4 | 21.2 | 19.3 | 34.6 |
| × | ✓ | × | × | 63.2 | 44.1 | 35.8 | 26.0 | 23.0 | 21.5 | 35.6 |
| ✓ | ✓ | × | × | 65.1 | 44.6 | 34.8 | 27.4 | 22.0 | 20.3 | 35.7 |
| ✓ | × | ✓ | × | 63.1 | 45.2 | 36.1 | 29.5 | 23.3 | 17.3 | 35.8 |
| ✓ | × | × | ✓ | 69.0 | 47.2 | 36.1 | 32.2 | 20.9 | 21.8 | 37.9 |
| × | ✓ | ✓ | × | 66.2 | 45.3 | 35.2 | 30.1 | 23.2 | 22.0 | 37.0 |
| × | ✓ | × | ✓ | 67.1 | 46.7 | 37.5 | 28.5 | 23.2 | 22.5 | 37.6 |
| ✓ | ✓ | ✓ | × | 68.8 | 46.8 | 36.3 | 27.8 | 21.8 | 21.5 | 37.2 |
| ✓ | ✓ | × | ✓ | 68.7 | 49.3 | 35.2 | 31.4 | 22.2 | 22.1 | 38.2 |
| ✓ | × | ✓ | ✓ | 70.3 | 49.1 | 35.8 | 32.5 | 22.3 | **22.7** | 38.8 |
| × | ✓ | ✓ | ✓ | 70.9 | 49.5 | 37.1 | 32.9 | 23.1 | 22.0 | 39.3 |
| ✓ | ✓ | ✓ | ✓ | **71.3** | **49.9** | **37.8** | **34.1** | **23.7** | 22.1 | **39.8** |

Table 6: Comparison results of generalization. The numbers in parentheses on the right represent the decrease in dAP values relative to those in the first row. The green font denotes the best performance. Under the 1/5/10-shot task setting, our method consistently achieves the lowest dAP values, demonstrating its strong generalization capability in addressing cross-domain tasks.

| Method | 1-shot FP↓ | 5-shot FP↓ | 10-shot FP↓ |
|--------|-----------|-----------|-------------|
| DINO DETR | 44.45 | 40.12 | 35.68 |
| DINOv2 | 29.73 (-14.72) | 27.70 (-12.42) | 30.26 (-5.42) |
| Ours | 26.47 (**-17.98**) | 22.54 (**-17.58**) | 17.98 (**-17.70**) |

mAP, demonstrating that the RR module effectively highlights foreground regions in the feature maps and filters out irrelevant background information. Using ER and RR together achieves the best result of 39.8 mAP, confirming their complementarity.

**Cross-Domain Generalization Analysis.** To validate that the VFMs have strong generalization and the detectors have poor generalization on the cross domain tasks, we fine-tune DINO DETR, DINOv2 and our method on six cross-domain datasets and evaluate their performance. DINOv2 refers to replacing the ResNet50 backbone of original DINO DETR with DINOv2. We use the average precision loss(dAP) caused by false positive samples as our evaluation metric. As shown in Table 6, DINO DETR has the highest dAP values reach 44.45/40.12/35.68 under the 1/5/10-shot task settings, respectively, indicating its poor generalization in cross domain tasks. In contrast, DINOv2 has lower dAP values of 29.73/27.7/30.26, indicating the strong generalization capability. Our method leverages a router module further to filter out irrelevant background region in VFM feature map. As a result, our method achieves the lowest dAP values of 26.47/22.54/17.98 under the 1/5/10-shot settings, respectively. Notably, under the 10-shot task setting, our method reduces dAP by 12.28 compared with DINOv2, validating the stronger generalization capability of our approach.

**Cross-Domain Localization Performance Analysis.** To validate that the detectors have strong localization capabilities and the VFMs lack accurate localization capabilities, we fine-tune DINO DETR, DINOv2 and our method on six cross-domain datasets and evaluate their performance under the 1-shot task setting. We use the decrease of AP75 relative to AP50 to evaluate model's localization capability. As shown in Table 7, DINOv2 shows the highest drop of 69.35%, indicating limited localization capabilities because its training dataset lacks accurate bounding box annotations. In comparison, DINO DETR exhibits better localization performance with a drop of 48.76%. Our method treats DINOv2 as an independent expert model to improve the detector's generalization rather than using DINOv2 directly as the detector backbone. Therefore, our method avoids the transfer of weak localization capabilities from the VFM to the detector. Experimental results show that our method achieves an attenuation of only 40.41%, demonstrating its effectiveness in maintaining the detector's localization performance.

## 4.2 Qualitative Visualizations

**Method Analysis.** To validate our method's effectiveness, we visualize the backbone feature maps. As shown in Figure 3, our method makes the model focus on foreground regions, showing that our router module can effectively filter out the useless background in the VFM feature maps. Additionally, we use t-SNE [80] to visualize backbone features on the DIOR dataset to demonstrate the model's

Table 7: Comparison results of localization capability under the 1-shot task setting. The arrows and values on the top-right denote the decrease in AP75 relative to AP50. Our method exhibits the smallest relative decrease, demonstrating strong robustness in localization performance.

| | DINO DETR | | DINOv2 | | Ours | |
|---|---|---|---|---|---|---|
| | AP50 | AP75 | AP50 | AP75 | AP50 | AP75 |
| | 12.51 | 6.41 ↓48.76% | 19.38 | 5.94 ↓69.35% | 28.48 | 16.97 ↓40.41% |

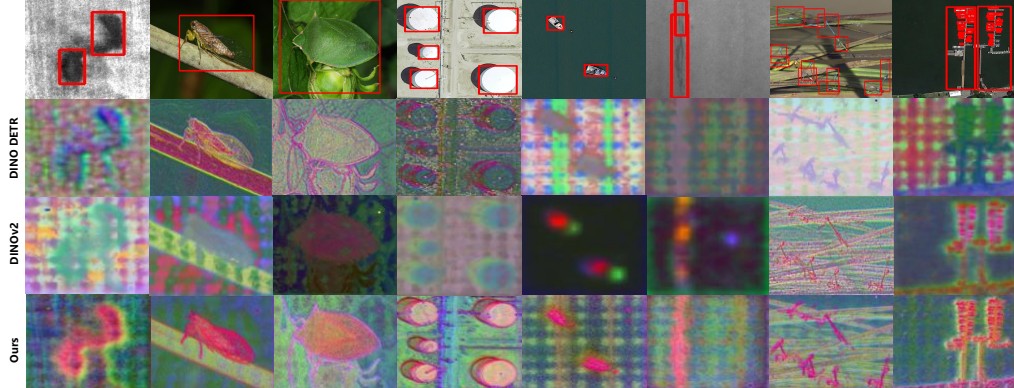

Figure 3: Visual comparison of backbone feature. The red regions in the figure represent show the model's focus regions.

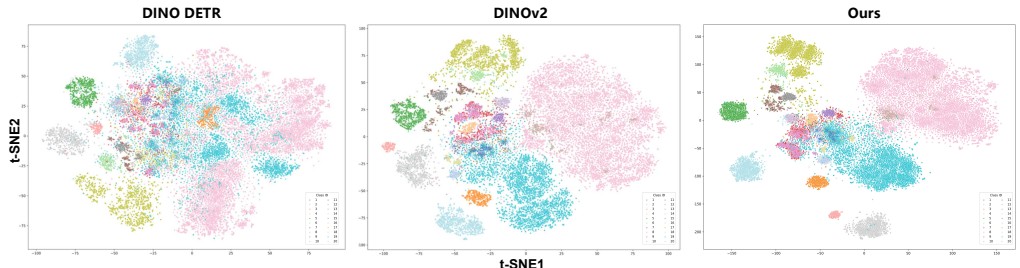

Figure 4: The t-SNE visualization of different detector features on the DIOR dataset. Each point in the figure represents a sample from the dataset, with color indicating its class. Our method effectively minimizes the distance between samples within the same class while maximizing the separation between samples of different classes.

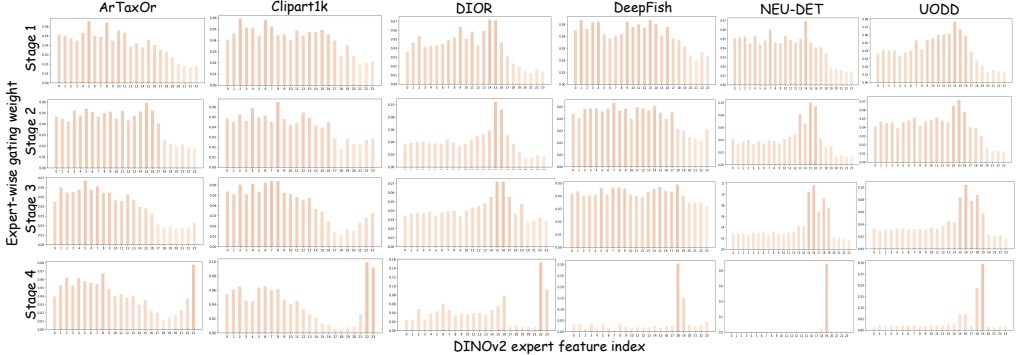

Figure 5: Expert feature selection at different layers of the backbone. The horizontal axis represents expert features, the vertical axis represents the expert-wise gating weights at different layers.

classification performance. As shown in Figure 4, our method shows higher intra-class compactness and inter-class separability, further demonstrating the strong generalization of our model.

**Expert-wise Routing Mechanism.** To illustrate that the importance of expert features varies across backbone layers, we analyze expert-wise gating weights. As shown in Figure 5, different backbone layers employ different VFM feature selection strategies. Deep detector features generally tend to

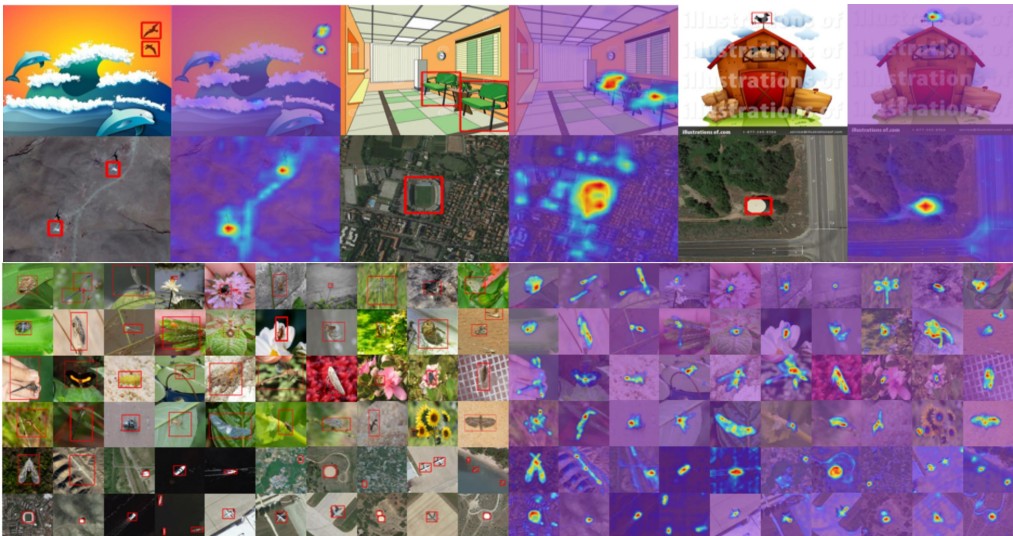

Figure 6: Heatmap visualization of region-wise gating weight, the brighter areas indicate higher levels of model attention.

select VFM expert feature from the deep VFM layer because they contain similar level image features. However, the deep detector features also select some shallow VFM expert features, showing that low-level image features, like object edges and colors, can also improve the high-level image feature. The VFM feature selection strategy at the same detector layer also changes on the different test datasets. These experimental results indicate that it is necessary to introduce a learnable module to dynamically adjust the VFM feature selection strategy. Therefore, inspired by the routing mechanism in MoE, we propose an expert-wise router method that generates expert gating weights based on detector features to select the VFM expert feature dynamically.

**Region-wise Routing Mechanism.** The information density is different in different regions of the VFM feature map. The foreground regions typically have higher information density, and background regions tend to have lower information density. To make the model to focus on the foreground region, we propose a region-wise router to filter background region of the VFM feature map and highlight foreground regions. To demonstrate the effectiveness of our method, we visualize the region-wise gating weights by using heatmap. As shown in Figure 6, after applying our region-wise router, the model focuses on foreground regions and ignores the irrelevant background regions.

## 5 Conclusion

In this paper, we propose a novel Cross-Domain Few-Shot Object Detection (CD-FSOD) paradigm leveraging a Mixture of Experts (MoE) architecture to integrate the vision foundation model with the well-trained detection model. Our method transfers VFM's generalization to the detector without modifying the original detector structure or retraining on the base class. We introduce expert-wise and region-wise routers to select VFM expert feature and filter irrelevant background regions in the VFM feature map. Additionally, we propose the shared expert projection module and private expert projections module, which decouple the shared and private image feature projections and minimize the parameters introduced by the VFM feature projection. Extensive experiments demonstrate the effectiveness of our approach in improving cross-domain performance.

## Acknowledgements

This work is supported in part by the National Natural Science Foundation of China (62192783, 62276128, 62406140), Young Elite Scientists Sponsorship Program by China Association for Science and Technology (2023QNRC001), the Key Research and Development Program of Jiangsu Province under Grant (BE2023019) and Jiangsu Natural Science Foundation under Grant (BK20221441, BK20241200). The authors would like to thank Huawei Ascend Cloud Ecological Development Project for the support of Ascend 910 processors.

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

# A  Ablation Study on Different Vision Foundation Models

To evaluate the impact of using different VFMs as expert models in our method, we compare the performance of using DINOv2 [27], CLIP [81], and combination of both as expert models. As shown in Table 8, DINOv2 has better performance than CLIP on cross-domain tasks. We further explore integrating multiple vision foundation models simultaneously to boost feature representation. The integration of DINOv2 and CLIP exceeds the GPU memory limitation, forcing us to reduce the batch size to 1. Experimental results demonstrate that integrating multiple vision foundation model leads to a slight performance improvement, confirming the complementary of different vision foundation models. However, this integration significantly increases training time and memory consumption. Considering the trade-off between performance improvement and computational costs, we ultimately only select DINOv2 as the expert model.

Table 8: Comparison results of combining different vision foundation models. The best results are highlighted with **bold**. *bs* denotes the batch size.

| Method | ArTaxOr | Clipart1k | DIOR | DeepFish | NEU-DET | UODD | Avg | Training time |
|---|---|---|---|---|---|---|---|---|
| Baseline (bs = 2) | 11.4 | 23.2 | 14.4 | 20.5 | 11.8 | 9.9 | 15.2 | 0.8h |
| + CLIP (bs = 2) | 53.7 | 47.7 | 35.0 | 28.9 | **24.0** | 19.2 | 34.8 | 1.7h |
| + DINOv2 (bs = 2, Ours) | 71.3 | 49.9 | 37.8 | **34.1** | 23.7 | **22.1** | 39.8 | 2.0h |
| + DINOv2&CLIP (bs = 1) | **71.8** | **50.2** | **38.1** | 33.9 | 23.8 | 21.7 | **39.9** | 3.6h |

# B  Ablation Study on DINOv2 Models of Different Parameter Sizes

To evaluate the impact of VFMs with different parameter scales, we use DINOv2-small, DINOv2-base, DINOv2-large, and DINOv2-giant as expert models and evaluate method permformance under the 10-shot task setting. As shown in Table 9, DINOv2-giant exceeds the memory capacity of our GPU, forcing us to reduce the batch size to 1 during training. The performance of DINOv2-giant is suboptimal due to its large intermediate feature maps, which require processing a large number of additional parameters. These extra parameters not only hinder effective fine-tuning on downstream tasks but also considerably increase training time. In contrast, **DINOv2-large offers a balanced trade-off between parameter size, training time, and performance improvement, delivering the best results within a reasonable computational cost.** Consequently, we select DINOv2-large as the expert model.

Table 9: Comparison results of DINOv2 Models of Different Parameter Sizes. *Size* denotes the parameter scales of different DINOV2 models.

| Method | ArTaxOr | Clipart1k | DIOR | DeepFish | NEU-DET | UODD | Training time | Size |
|---|---|---|---|---|---|---|---|---|
| DINOv2-S (bs = 2) | 54.9 | 38.5 | 30.4 | 28.5 | 22.4 | 20.2 | 1.1h | 21M |
| DINOv2-B (bs = 2) | 67.5 | 45.9 | 35.6 | 30.4 | 21.6 | 21.3 | 1.5h | 86M |
| DINOv2-L (bs = 2, Ours) | **71.3** | **49.9** | **37.8** | **34.1** | 23.7 | **22.1** | 2.0h | 300M |
| DINOv2-g (bs = 1) | 69.1 | 48.7 | 36.7 | 33.1 | 22.0 | 21.4 | 4.5h | 1100M |

# C  Ablation Study on Hyperparameters Setting

In the shared and private expert projection modules, we introduce the hyperparameters $m$, to control the parameter scales of the shared expert projection module and the private expert projection module. *e.g.*, $\boldsymbol{F}_s^i = \boldsymbol{F}_D^i \cdot \boldsymbol{\theta}_s \in \mathbb{R}^{B \times C_s \times H \times W}$, $\boldsymbol{F}_p^i = \boldsymbol{F}_d^i \cdot \boldsymbol{\theta}_p^i \in \mathbb{R}^{B \times C_p \times H \times W}$, where $C_s = \frac{m-1}{m}C, C_p = \frac{1}{m}C$. To determine the optimal configuration, we conduct an ablation study on the values of m. As shown in Table 10, our method shows the best performance when $m = 16$.

In the mixture of experts module, we introduce two hyperparameters, $\alpha$ and $\beta$, *e.g.*, $\boldsymbol{F}_f^l = \boldsymbol{F}^l + \sum_{n=1}^{N} \left( \alpha \cdot \boldsymbol{G}_e^{n,l} \circledast \boldsymbol{F}_{d'}^{n,l} + \beta \cdot \boldsymbol{G}_r^{n,l} \circledast \boldsymbol{F}_{d'}^{n,l} \right)$. To evaluate the impact of these parameters, we perform an ablation study on $\alpha$ and $\beta$. As shown in Table 11, the results indicate that the optimal configuration is $\alpha = 0.5$ and $\beta = 0.5$.

Table 10: The 10-shot ablation results on hyperparameters $m$ and $n$.

| m | ArTaxOr | Clipart1k | DIOR | DeepFish | NEU-DET | UODD | Avg |
|---|---------|-----------|------|----------|---------|------|-----|
| 2 | **71.9** | 49.6 | 36.6 | 31.1 | 22.8 | 18.2 | 38.4 |
| 4 | 70.0 | 49.0 | 37.3 | 32.5 | 21.1 | 21.4 | 38.6 |
| 8 | 70.3 | 48.9 | 35.7 | 32.3 | 22.4 | 21.4 | 38.5 |
| 16 (Ours) | 71.3 | **49.9** | **37.8** | **34.1** | **23.7** | 22.1 | **39.8** |
| 32 | 69.5 | 48.5 | 36.4 | 33.2 | 23.1 | **22.9** | 38.9 |

Table 11: The 10-shot ablation results on hyperparameters $\alpha$ and $\beta$.

| $\alpha$ & $\beta$ | ArTaxOr | Clipart1k | DIOR | DeepFish | NEU-DET | UODD | Avg |
|---------------------|---------|-----------|------|----------|---------|------|-----|
| 0.1, 0.9 | 71.1 | 49.1 | 35.3 | **34.9** | 22.9 | **22.8** | 39.4 |
| 0.3, 0.7 | 70.5 | 49.7 | 30.5 | 34.1 | 23.0 | 20.6 | 38.1 |
| 0.5, 0.5 (Ours) | **71.3** | **49.9** | **37.8** | 34.1 | **23.7** | 22.1 | **39.8** |
| 0.7, 0.3 | 70.1 | 49.2 | 37.6 | 31.9 | 21.7 | 18.1 | 38.1 |
| 0.9, 0.1 | 68.2 | 49.8 | 36.4 | 32.2 | 21.1 | 21.0 | 38.1 |

# D  Ablation Study on Fine-Tuning Strategy

To evaluate the impact of different fine-tuning strategies, we evaluate the performance of four fine-tuning strategies: Full Finetune, LoRA [82], Partial Finetune and our fine-tuning strategy. As shown in Table 12, All fine-tuning Strategies keep the vision foundation model frozen without updating its parameters. LoRA-based fine-tuning achieves the lowest number of trainable parameters but suffers from limited performance. Partially fine-tuning the model while excluding the backbone introduces a moderate increase in trainable parameters and yields improved results. Full fine-tuning achieves the highest accuracy, yet comes at the cost of significant training overhead and a heightened risk of overfitting in few-shot scenarios. In contrast, our proposed strategy, which fine-tunes only the classification head, regression head and the new proposed module, strikes an effective balance between computational efficiency and detection performance, demonstrating strong generalization in cross-domain few-shot object detection tasks.

Table 12: The 10-shot ablation results on different finetuning strategies. *Full Finetune* denotes fine-tuning all model parameters. *LoRA* denotes fine-tuning all model based on Low-Rank Adaptation. *Partial Finetune* denotes fine-tuning all components except the backbone. *Ours* denotes fine-tuning only the classification head, regression head, and the proposed module. *Train params* indicates the number of trainable parameters.

| Fine-Tuning Strategy | ArTaxOr | Clipart1k | DIOR | DeepFish | NEU-DET | UODD | Avg |
|----------------------|---------|-----------|------|----------|---------|------|-----|
| Full Finetune | 55.2 | 39.0 | 32.6 | 22.3 | 20.8 | 19.3 | 31.5 |
| LoRA | 44.5 | 28.0 | 26.7 | 20.9 | 18.4 | 15.9 | 25.7 |
| Partial Finetune | 50.7 | 34.7 | 30.9 | 22.6 | 20.9 | 16.2 | 29.3 |
| Ours | **71.3** | **49.9** | **37.8** | **34.1** | **23.7** | **22.1** | **39.8** |

# E  Ablation Study on Feature Aggregation Strategy

To validate the effectiveness and necessity of our proposed MoE-based feature aggregation strategy, we compare our method with four simpler feature aggregation strategies. Specifically, we design and evaluated four variants:

- **All Stages**: Retain all stage PEP modules and replace them with simple MLPs, removing all other modules.

- **Final Stage**: Retain only the final-stage PEP module and replace it with a simple MLP, removing all other modules.

- **Linearly Adding**: Remove all modules and aggregate features at the backbone stage via linear addition.

- **Learnable Weight**: Introduce a learnable weight parameter to aggregate features from different layers of the VFM.

As shown in Table 13, simple feature aggregation strategies lead to a substantial performance decline, thereby validating the efficacy of our approach. Learnable Weight feature aggregation strategy introduces learnable weight parameters as a simple routing network to select and aggregate VFM features, thereby maintaining the MoE structure similar to ours. In contrast, the other three aggregation strategies (All Stages, Final Stage, Linearly Adding) simply aggregate the VFM features into the backbone features without selecting them. Therefore, it outperforms the other simple aggregation strategies. However, it still leads to a decline in performance compared to our method, highlighting the importance of backbone features in guiding VFM feature selection.

Table 13: Comparison results of our method and simple feature aggregation strategies.

| Method | ArTaxOr | Clipart1k | DIOR | DeepFish | NEU-DET | UODD | Avg |
|---|---|---|---|---|---|---|---|
| All Stages | 56.3 | 44.4 | 34.1 | 26.4 | 16.8 | 17.6 | 32.6 |
| Final Stage | 48.8 | 43.8 | 30.7 | 24.1 | 17.1 | 17.4 | 30.3 |
| Linearly Adding | 2.5 | 5.1 | 9.1 | 4.7 | 3.9 | 5.4 | 5.1 |
| Learnable Weight | 62.5 | 47.4 | 31.6 | 31.1 | 19.1 | 20.7 | 35.4 |
| Ours | **71.3** | **49.9** | **37.8** | **34.1** | **23.7** | **22.1** | **39.8** |

# F  Comparison with Recent Works

**Comparison with ETS.** We compare our method with ETS [56] on six cross domain datasets. For fairness, we reproduce the experiments using the official ETS code. As shown in Table 14, our method can be directly applied to Grounding DINO without any modification and achieves similar performance to ETS, even though it is not specifically designed for Grounding DINO.

Table 14: Comparison results of ETS method and our method under the 10-shot task setting.

| Method | ArTaxOr | Clipart1k | DIOR | DeepFish | NEU-DET | UODD | Avg |
|---|---|---|---|---|---|---|---|
| ETS (Our implementation) | 70.6 | **60.8** | 37.5 | **42.8** | 26.1 | 28.3 | 44.4 |
| Grounding DINO (Swin-B) + Ours | **73.8** | 60.3 | **40.2** | 37.0 | **27.9** | **29.2** | **44.7** |

**Comparison with Frozen DETR.** We compare our method with Frozen DETR [83]. We adopt the same baseline structure as Frozen DETR. Frozen DETR follows a two-stage training paradigm of base training and finetuning. To align the experimental and dataset settings, we skip the base training step, finetuning only on the cross domain datasets. To validate the effectiveness of the method on in-domain datasets, we train our method on COCO for 12 epochs and compare it to the performance of Frozen DETR reported in the original paper. As shown in Table 15, our method consistently outperforms Frozen DETR across six cross domain datasets and the COCO dataset.

Table 15: Comparison results of Frozen DETR and our method under the 10-shot task setting.

| Method | ArTaxOr | Clipart1k | DIOR | DeepFish | NEU-DET | UODD | COCO |
|---|---|---|---|---|---|---|---|
| DINO(ResNet50) | 2.9 | 13.6 | 6.9 | 11.6 | 4.5 | 2.8 | 49.0 |
| Frozen DETR | 45.8 | 33.9 | 31.5 | 17.5 | 9.1 | 3.2 | 51.9 |
| DINO(ResNet50) + Ours | **71.3** | **49.9** | **37.8** | **34.1** | **23.7** | **22.1** | **55.2** |

# G  Analysis of Routing Network Based on Detector Feature

To validate the feasibility of using detector features to route VFM features, we visualize the feature maps of detector and VFM using PCA. As shown in Figure 7, the detector shows higher attention in the foreground regions, reflecting its strong localization ability. However, the foreground object feature are unclear, leading to weaker classification performance. In contrast, the VFM extracts clear feature representations but maintains high attention in multiple background regions, indicating its weaker localization ability. Therefore, we leverage the detector's localization ability to guide VFM feature maps in supplementing the missing key features in foreground regions.

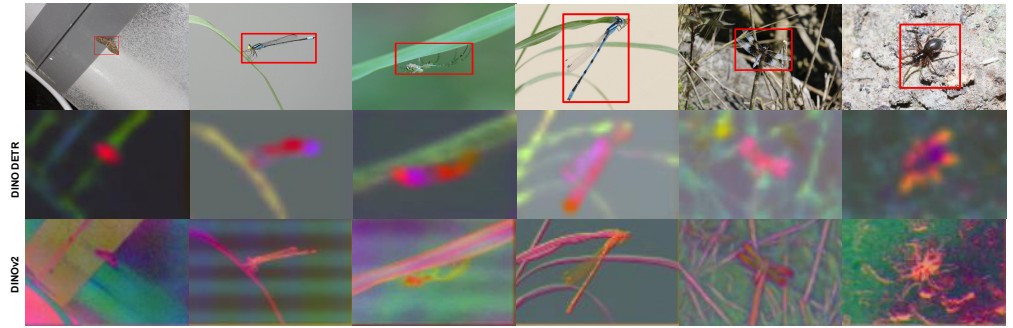

Figure 7: Visual comparison of detector feature and DINOv2 feature. The red regions in the figure represent the areas of focus during feature extraction by the model.

