# OpenReview forum: "DON’T NEED RETRAINING: A Mixture of DETR and Vision Foundation Models for Cross-Domain Few-Shot Object Detection"
_NeurIPS.cc/2025/Conference — NeurIPS 2025 poster_

### Official Review · Reviewer_KWMR · 2025-06-29

**Clarity:** 3
**Significance:** 4
**Originality:** 4
**Rating:** 5
**Confidence:** 5

**Summary:**

The paper proposes incorporating the general representation capabilities of pre-trained vision foundation models as expert knowledge into object detectors, aiming to enhance their detection performance in unseen domains. Specifically, the output features from each layer of the VFM are treated as individual experts. A Mixture-of-Experts-based approach is introduced to dynamically route and modulate the VFM features. Experiments demonstrate that the proposed method achieves performance improvements in CD-FSOD tasks and can be used as a plug-in module for various detectors.

**Questions:**

Please answer the following questions:
1. Addressing the ambiguities noted in Weakness 1.
2. Please compare with Frozen-DETR on the CD-FSOD datasets.
3. Is the proposed method effective on in-domain datasets with sufficient samples, such as COCO? Please supplement the experiments of the proposed method on the COCO dataset and compare the results with Frozen-DETR.

Since the motivation of this paper is similar to that of Frozen-DETR, I am very concerned about the comparison and discussion between the proposed method and Frozen-DETR. This will affect my decision on adjusting the score.

In addition, I have the following suggestion:
In Table 1, † indicates that the methods were fine-tuned on the six datasets. In fact, Distill-cdfsod, CD-ViTO, and the proposed method (Ours) all require fine-tuning. The authors are advised to revise this description to avoid misunderstandings.

**Ethical Concerns:**

["NO or VERY MINOR ethics concerns only"]

**Final Justification:**

The authors have addressed all my concerns. In particular, the comparison with Frozen DETR demonstrates the advantage of using the proposed MoE-based fusion of VFM features over direct fusion. The results on the COCO dataset show that the proposed method not only improves performance in cross-domain few-shot tasks but also enhances general object detection.

Overall, I have decided to raise my final rating to 5.

**Limitations:**

Limitations are not mentioned.

**Paper Formatting Concerns:**

No formatting issues observed.

**Quality:**

4

**Strengths And Weaknesses:**

Strengths:
1. The motivation is clear and meaningful.
2. The proposed method is simple and effective, and it can be extended to various DETR-style detectors.
3. The paper provides comprehensive analytical experiments.

Weaknesses:
1. The paper contains some unclear parts. For example, (1) The feature shapes of DINOv2 differ from those of the backbone (ResNet50) in DINO at various stages, yet the paper uniformly represents them using H×W, which may cause confusion. (2) According to Fig. 5, does the proposed method use all 24 output features from DINOv2-L as expert features but perform feature fusion only at the output features of the four stages of ResNet50? (3) What does DINOv2 refer to in Tables 4 and 5, and how does it work?
2. There is a lack of comparison with related work. The motivation of the paper is similar to that of Frozen-DETR [1], which also transfers the general representations extracted by the frozen VFM to object detectors. It is recommended that the authors include a comparison with Frozen-DETR.
3. The method proposed in this paper has been evaluated solely on the CD-FSOD datasets. Its effectiveness on general object detection tasks has yet to be demonstrated.

[1] Fu S, Yan J, Yang Q, et al. Frozen-DETR: Enhancing DETR with Image Understanding from Frozen Foundation Models[J]. arXiv preprint arXiv:2410.19635, 2024.

---

> ### Author Rebuttal · Authors · 2025-07-31
>
> We sincerely thank Reviewer KWMR for the constructive feedback, insightful questions, and valuable suggestions!
>
> Below we address your concerns.
>
> >**Q1: About some unclear parts in the paper.**
>
> **R1**: Thank you for your valuable questions.
>
> Below is our detailed response to the three issues you raised in Question 1:
>
> + R1.1: In DINOv2, the spatial dimensions of the feature maps differ from those of the ResNet backbone’s feature maps. In the revised main paper., we will use (H, W) to denote the spatial dimensions of backbone features and (Hd, Wd) to denote the spatial dimensions of DINOv2 features.
> We will revise this phrasing in the revised main paper.
>
> + R1.2: Our feature fusion strategy follows the illustration: We first process the output of the previous stage with the backbone layer. The processed features are then fused with the VFM features. Finally, the fused features are used as the input for the next layer.
>
>
> + R1.3: In Tables 4 and 5, “DINO v2” refers to replacing the ResNet50 backbone of original DINO DETR  with DINO v2, while keeping the detection head and training settings unchanged. This ensures a fair experimental comparison.
>
> >**Q2: Comparison with Frozen DETR on cross‑domain datasets.**
>
> **R2**: Thank you for bringing Frozen DETR [1] to our attention.
>
> We compare our method with Frozen DETR. We adopt the same baseline structure as Frozen DETR. Frozen DETR follows a two‑stage training paradigm of base training and finetuning. To align the experimental and dataset settings, we skip the base training step, finetuning only on the cross domain datasets. As shown in Table 1, our method consistently outperforms Frozen DETR across six crossdomain datasets. We will include these comparison experiments in the revised main paper.
>
>
> **Table 1: Comparison Results Against Frozen DETR under 10-shot setting.**
>
> | Method                | ArTaxOr | Clipart1k |  DIOR  | DeepFish | NEU-DET |  UODD  |Avg|
> |:----------------------|:-------:|:---------:|:------:|:--------:|:-------:|:------:|:------:|
> | Frozen DETR   |  45.8   |    33.9    |  31.5   |   17.5    |   9.1   |  3.2   | 23.5|
> | Ours     |  71.3   |   49.9    |  37.8   |  34.1    |   23.7   |  22.1   |39.8|
>
> **Discussion between our method and Frozen DETR.**
>
> +  Frozen DETR modifies the original DINO DETR architecture and introduces many additional parameters, thereby requiring base training followed by fine-tuning on novel classes to achieve strong performance. In contrast, our approach achieves best performance by fine-tuning only on novel classes, without any base training.
> + Frozen DETR enhances detector features using only the final‑layer output of the VFM. Because it relies on self‑attention for feature enhancement, it incurs significant computational overhead. In contrast, our method employs the SEP and PEP modules, avoiding the \(O(n^2)\) complexity, which allows us to fuse multi‑layer VFM features efficiently.
>
> References:
>
> [1] Fu S, Yan J, Yang Q, et al. Frozen-DETR: Enhancing DETR with Image Understanding from Frozen Foundation Models[J]. arXiv preprint arXiv:2410.19635, 2024.
>
> >**Q3: Comparison with FrozenDETR on the COCO dataset.**
>
> **R3**: Thank you for your constructive suggestion.
>
> We train and evaluate our method on the COCO dataset and compared it to the performance of Frozen DETR reported in the original paper. As shown in Table 2, our method outperforms Frozen DETR on COCO, further validating its general applicability. We will include these comparison experiments in the revised supplementary material.
>
>
> **Table 2: Comparison Results Against Frozen DETR on the COCO dataset.**
>
> | Method                | Backbone | Epochs |  AP  |
> |:----------------------|:-------:|:---------:|:------:|
> | DINO DETR              |  R50   |    12    |  49.0   |
> | Frozen DETR  |  R50   |    12    |  51.9   |
> |  DINO DETR + Ours  |  R50   |    12    |  55.2   |
>
>
> >**Q4: Analysis of method limitations.**
>
> **R4**: Thanks for your insightful comment.
>
> One limitation of our approach is that it is currently confined to visual tasks and does not yet support multimodal applications such as text‑guided visual grounding or video synthesis. However, our enhancement strategy can be easily applied to the visual backbones of multimodal models. As shown in Table 3, the method achieves strong performance when combined with Grounding DINO, even without explicit optimization for textual features. Extending the approach to multimodal scenarios will be explored in future work.
>
> **Table 3: Comparison results of original Grounding DINO and our method under 10-shot setting.**
>
> | Method                  | ArTaxOr | Clipart1k |  DIOR  | DeepFish | NEU-DET |  UODD  | Avg|
> |:----------------------|:-------:|:---------:|:------:|:--------:|:-------:|:------:|:------:|
> | Grounding DINO (Swin-B)   |  72.0   |    58.4   |  38.2   |   35.2    |   26.5   |  28.6   |43.2|
> | Grounding DINO (Swin-B) + Ours   |  73.8   |    60.3    |  40.2   |   37.0   |   27.9   |  29.2   |44.7|
>
> >**Q5: About the symbol used in Table 1.**
>
> **R5:** Thank you for your constructive suggestion.
>
> Distill‑CDFSOD, CD‑ViTO, and our method all need to finetune on novel classes. We will revise this phrasing in the revised main paper.

---

> > ### Comment · Area_Chair_Edbb · 2025-08-04
> > **URGENT: Discussion**
> >
> > Dear reviewer,
> >
> > Please engage in discussion.
> >
> > regards
> >
> > AC

---

> ### Comment · Reviewer_KWMR · 2025-08-05
>
> Thanks for the authors' response. And my question has been resolved.
>
> **Q1: About some unclear parts in the paper.**
>
> Regarding the unclear parts in the paper, I strongly recommend that the authors release their complete code after the paper is accepted.
>
> **Q2 & Q3: Comparison with Frozen DETR.**
>
> First, I would like to emphasize that Frozen DETR, just like the method proposed in this paper, requires both base training (e.g., on COCO) and novel fine-tuning (on cross-domain datasets). Therefore, the authors should not highlight that their method skips base training.
>
> Secondly, the result presented in Table 2 of R3, on the COCO dataset, are surprising to me. It indicates that incorporating features from vision foundation models can significantly improve object detection performance. I suggest the authors include this part in the appendix of the paper.

---

> ### Author Response · Authors · 2025-08-06
> **Very happy that we addressed your concerns! Thanks for your insightful and valuable suggestions!**
>
> **Thank you for your positive feedback!**
>
> Below we address your concerns.
>
> >**Q1: About code.**
>
> **R1:** Thank you for your valuable suggestions.
>
> We will release our complete code if the paper is accepted.
>
> >**Q2: Comparison of our method and Frozen DETR.**
>
> **R2:** Thank you for your insightful comment.
>
> As shown in Table 1, a key difference is that our method can directly use DETR checkpoint pretrained on COCO and finetune the added modules  on novel classes, making it simple and efficient to apply. In contrast, Frozen DETR modifies the original detector architecture and cannot directly make use of pretrained DETR weights, so it requires training a new model from scratch. We will include relevant discussions and comparison results with Frozen DETR on six cross-domain datasets in the revised supplementary material.
>
> **Table 1: Method comparison of our method and Frozen DETR.**
>
> |            | Frozen DETR | Our Method |
> |:-----------------|:-----------|:----------|
> | Newly Added Modules | Trained on massive base data | Only trained on small-scale novel data |
> | DETR Architecture | Modify the DETR (change channel dimensions of the self-attention modules) | Retain the DETR original architecture |
> | DETR Weights | Jointly trained with the added modules on base data | Retain the DETR original weights |
>
> >**Q3: Experiment results on COCO dataset.**
>
> **R3:** Thank you for your recognition of our work.
>
> We will include comparison results with Frozen DETR on COCO dataset in the revised supplementary material.

---

### Official Review · Reviewer_Xh4w · 2025-07-01

**Clarity:** 4
**Significance:** 4
**Originality:** 3
**Rating:** 5
**Confidence:** 5

**Summary:**

This paper introduces a cross-domain few-shot object detection method that does not require retraining on the source domain. It directly employs a mixture-of-experts (MoE) architecture, where a visual foundation model (DINOv2) provides expert features, and an object detector (DINO) serves as the router network, complementing each other in terms of generalization and localization capabilities. The router network includes expert-wise and region-wise modules that dynamically and iteratively aggregate coarse-grained and fine-grained visual information from the VFM into the object detector's backbone features at each stage. Expert features are projected from the VFM feature maps through shared and private projection modules, effectively modeling both common and expert-specific knowledge. Experiments show that the proposed method outperforms prior approaches on the CD-FSOD benchmark, demonstrates strong transferability across different object detection backbones, and validates the effectiveness of the introduced router and projection modules through ablation studies.

**Questions:**

Aside from the questions raised in the above section, I'd like to gain a deeper understanding of the intuition behind this architectural design. Fundamentally, why can an OD that's sensitive to domain shift reliably route VFM features? I know the authors show the effectiveness of the method through experiments, but theoretically, using the OD features to route VFM features seems to be counterintuitive and ill-posed. Would like to see further clarification/elaboration on this design in the paper.

Also, it would be great to hear authors' thoughst on why for DINOv2, dAP increase to 30.26 from 27.7 , for 5-shot to 10-shot according to the Table 4.

**Ethical Concerns:**

["NO or VERY MINOR ethics concerns only"]

**Final Justification:**

The authors ran additional experiments showing the effectiveness of all modules introduced, further analyzed the experiments, and explained the rationale behind the design. My latest comment is intended to encourage the authors to further improve the completeness of the paper. Overall, this work is technically solid and worth more attention in this field, so I would increase the score to 5.

**Limitations:**

The authors didn't explicitly list the limitations; it would be helpful for future work in this direction if they could share any limitations they're aware of.

**Paper Formatting Concerns:**

No outstanding formatting concerns.

**Quality:**

3

**Strengths And Weaknesses:**

Strengths:
1) The motivation is clear: VFMs pretrained on large-scale data exhibit better generalization but lack accurate localization capabilities, while object detectors (OD) specialize in bounding box regression but are less robust to domain shifts.
2) The design is straightforward yet effective, leveraging pretrained models without the need for retraining on base classes. This brings practical value to real-world scenarios, where downstream applications demand quick and performant adaptation.
3) The method exhibits significant performance advantages over existing approaches on the CD-FSOD benchmark. It also showcases transferability across a variety of DETR architectures.


Weaknesses:
1. Clarity of ablation studies: It is unclear what it means for SEP and PEP to exist without ER and RR. How are the VFM features aggregated in this case? Is there still a gating network, or are the features aggregated in a way that is not conditioned on the backbone features?

2. Comprehensiveness of ablation studies: What is the rationale for not enumerating all possible combinations? Since the projection methods and router mechanisms are orthogonal, there should be 3 (SEP only, PEP only, SEP+PEP) × 4 (ER + RR, ER only, RR only, neither) + 1 (DINO only) = 13 combinations.

3. Lack of simpler baselines: Additional baselines are needed to show the necessity of the MoE structure. Table 3 shows a significant performance boost from DINO to SEP only, suggesting that much of the improvement may stem from the rich visual features provided by DINOv2. Did the authors explore simpler baselines, such as aggregating VFM features using a simple MLP (without conditioning on backbone features), or combining VFM features only at the final stage with the backbone features to justify the iterative design?

4. Notations to be improved/typos to be corrected:

    a) At the start of section 3.4, n seems to be overloaded, can be number of experts, also said to be a hyperpamter in expert   projection module;

    b) F_D^l’, prime looks like indicate another layer l’, which I believe the intention here is still to indicate layer l;

    c) Section 4.1, text description is not consistent with the table. Shouldn’t it be 24.6 mAP improvement on 10-shot for the     proposed method vs  DINO. Shouldn’t it be 9.8 improvement on 10-shot for the proposed method vs CD-ViTO

5. You may want to cite some more recent work:

     [1] Pan, Jiancheng, et al. "Enhance then search: An augmentation-search strategy with foundation models for cross-domain few-shot object detection." Proceedings of the Computer Vision and Pattern Recognition Conference. 2025.

     [2]Meng, Boyuan, et al. "CDFormer: Cross-Domain Few-Shot Object Detection Transformer Against Feature Confusion." arXiv preprint arXiv:2505.00938 (2025).

     [3] Huang, Yali, et al. "Instance Feature Caching for Cross-Domain Few-Shot Object Detection." Proceedings of the Computer Vision and Pattern Recognition Conference. 2025.

---

> ### Author Rebuttal · Authors · 2025-07-31
>
> We sincerely thank Reviewer Xh4w for the constructive feedback, insightful questions, and valuable suggestions!
>
> Below we address your concerns.
>
> >**Q1: Clarification of VFM feature aggregation without ER and RR.**
>
> **R1**: Thank you for your insightful comment.
>
> When the ER and RR modules are removed, we assign a unit weight to each VFM feature map. We then directly apply the SEP and PEP modules to all VFM features maps without backbone‑conditioned guidance or additional gating network.
>
>
> >**Q2: Additional ablation studies.**
>
> **R2**:   Thank you for your careful attention to our ablation studies.
>
> We conduct comprehensive ablation studies, as shown in Table 1. We will include comprehensive ablation studies in the revised main paper.
>
> **Table 1: Results of ablation studies. 'SEP' denotes shared expert projection，'PEP' denotes the private expert projection，'ER' denotes the expert-wise router，'RR' denotes the region-wise router.**
>
> |  SEP  |  PEP  |   ER   |  RR   | ArTaxOr | Clipart1k |  DIOR   | DeepFish | NEU-DET |  UODD  | Avg |
> |:-----:|:-----:|:------:|:-----:|:-------:|:---------:|:-------:|:--------:|:-------:|:------:|:------:|
> |   ×   |   ×   |   ×    |   ×   |  11.4   |   23.2    |  14.4   |  20.5    |  11.8   |  9.9   | 15.2 |
> |   √   |   ×   |   ×    |   ×   |  62.1   |   43.8    |  34.5   |  26.4    |  21.2   | 19.3   | 34.6 |
> |   ×   |   √   |   ×    |   ×   |  63.2   |   44.1    |  35.8   |  26.0    |  23.0   | 21.5   | 35.6 |
> |   √   |   √   |   ×    |   ×   |  65.1   |   44.6    |  34.8   |  27.4    |  22.0   | 20.3   | 35.7 |
> |   √   |   ×   |   √    |   ×   |  63.1   |   45.2    |  36.1   |  29.5    |  23.3   | 17.3   | 35.8 |
> |   √   |   ×   |   ×    |   √   |  69.0   |   47.2    |  36.1   |  32.2    |  20.9   | 21.8   | 37.9 |
> |   ×   |   √   |   √    |   ×   |  66.2   |   45.3    |  35.2   |  30.1    |  23.2   | 22.0   | 37.0 |
> |   ×   |   √   |   ×    |   √   |  67.1   |   46.7    |  37.5   |  28.5    |  23.2   | 22.5   | 37.6 |
> |   √   |   √   |   √    |   ×   |  68.8   |   46.8    |  36.3   |  27.8    |  21.8   | 21.5   | 37.2 |
> |   √   |   √   |   ×    |   √   |  68.7   |   49.3    |  35.2   |  31.4    |  22.2   | 22.1   | 38.2 |
> |   √   |   ×   |   √    |   √   |  70.3   |   49.1    |  35.8   |  32.5    |  22.3   | 22.7   | 38.8 |
> |   ×   |   √   |   √    |   √   |  70.9  |   49.5    |  37.1   |  32.9    |  23.1   | 22.0   | 39.3|
> |   √   |   √   |   √    |   √   |  71.3   |   49.9    |  37.8   |  34.1    |  23.7   | 22.1   | 39.8 |
>
> >**Q3: Evaluation of simpler feature aggregation strategy.**
>
> **R3**: Thank you for your valuable suggestions.
>
> We compare our method to three simpler feature aggregation strategies. Specifically, we designed and evaluated three variants:
>
> + All Stages: Retain all stage PEP modules and replace them with simple MLPs, removing all other modules.
>
> + Final Stage: Retain only the final-stage PEP module and replace it with simple MLP, removing all other modules.
>
> + Linearly Adding: Remove all modules and aggregate features at the backbone stage via linear addition,
>
>  As shown in Table 2, simple feature aggregation strategies lead to a substantial performance decline, thereby validating the efficacy of our approach. Moreover, aggregating features only at the final backbone stage performs worse than aggregating at every stage, highlighting the importance of our iterative design. We will include these comparison experiments in the revised supplementary material.
>
> **Table 2: Comparison Results of our method and simple feature aggregation strategies.**
>
> | Method                  | ArTaxOr | Clipart1k |  DIOR  | DeepFish | NEU-DET |  UODD  | Avg |
> |:----------------------|:-------:|:---------:|:------:|:--------:|:-------:|:------:|:------:|
> | All Stages    |  56.3  |   44.4      |  34.1   |   26.4    |   16.8   |  17.6   |32.6|
> | Final Stage      |   48.8    |  43.8   |  30.7   |  24.1    |   17.1   |  17.4   |30.3|
> | Linearly Adding    |  2.5   |   5.1    |  9.1   |  4.7   |   3.9   | 5.4   |5.1|
> | Ours   |  71.3   |   49.9    |  37.8   |  34.1    |  23.7   | 22.1   |39.8|
>
> >**Q4: About recent works.**
>
> **R4:** Thank you for pointing us to [1] ,[2] and [3].
>
> We acknowledge their relevance to our work and will cite these papers in the revised main paper.
>
>
> ```
> ETS[1] proposes a two-step augmentation and grid-search strategy to adapt foundation models for cross-domain few-shot object detection. CDFormer[2] introduces transformer-based OBD and OOD modules to mitigate object-background and inter-class feature confusion in cross-domain few-shot detection. IFC[3] enhances cross-domain few-shot detection by using learnable instance feature caches with an instance reweighting module for robust prototypes.
> ```
>
> References:
>
> [1] Pan, Jiancheng, et al. "Enhance then search: An augmentation-search strategy with foundation models for cross-domain few-shot object detection." Proceedings of the Computer Vision and Pattern Recognition Conference. 2025.
>
> [2]Meng, Boyuan, et al. "CDFormer: Cross-Domain Few-Shot Object Detection Transformer Against Feature Confusion." arXiv preprint arXiv:2505.00938 (2025).
>
> [3] Huang, Yali, et al. "Instance Feature Caching for Cross-Domain Few-Shot Object Detection." Proceedings of the Computer Vision and Pattern Recognition Conference. 2025.
>
> >**Q5: About notations and typos.**
>
> **R5:** Thank you for your careful attention to our paper.
>
> We have carefully revised all the writing issues you pointed out and re‑checked the entire paper to ensure that similar mistakes will not occur again.
>
>
> >**Q6: Analysis of routing network based on the OD feature.**
>
> **R6**: Thank you for your insightful question.
>
> We empirically observe that detectors possess strong localization and regression capabilities, accurately identifying object regions. However, due to domain shift, object features within these regions are often incomplete. This aligns with our paper’s conclusion that detectors have strong localization but weak classification. Therefore, we leverage the detector’s localization ability to guide VFM feature maps in supplementing the missing key features in salient regions so that the more generalizable VFM features can compensate for the information loss caused by domain shift. We will include the corresponding visualizations and discussions in the revised supplementary material.
>
> >**Q7: Analysis of abnormal variations in DINOv2’s classification capability.**
>
> **R7**: Thank you for your constructive feedback.
>
> We argue that the increased sample size leads the model to prioritize localization regression at the expense of its classification accuracy under the 10‑shot task setting. As shown in table 3, We observe that although DINOv2 exhibits strong classification performance(dAP: 27.7) in the 5‑shot setting, its average performance(mAP: 23.3) is low. In contrast, despite weaker classification performance(dAP: 30.3), the average performance(mAP: 27.4) increases substantially in the 10‑shot setting, supporting our conclusion.
>
> **Table 3: Comparison Results of DINOv2 under 5/10 shot task settings.**
>
> | Model                  | ArTaxOr | Clipart1k |  DIOR  | DeepFish | NEU-DET |  UODD  | Avg |
> |:----------------------|:-------:|:---------:|:------:|:--------:|:-------:|:------:|:------:|
> | DINOv2(5-shot)   |  48.3  |   29.5      |  22.4   |   19.7    |   10.5   |  9.3   |23.3|
> | DINOv2(10-shot)    |   51.9    |  34.2   |  25.6   |  24.1    |   15.2   |  13.5   |27.4|
>
>
>
>
> >**Q8: Analysis of method limitations.**
>
> **R8:** Thank you for your constructive suggestion.
>
> One major limitation of our approach is that it currently focuses solely on visual tasks and has not been extended to multimodal applications such as text‑guided visual grounding or video synthesis. Nevertheless, the proposed enhancement strategy can be easily applied to the visual backbones of multimodal models. As demonstrated in Table 4, our method still performs strongly when combined with Grounding‑DINO, even without explicit optimization for textual features. We intend to further explore this direction in future work.
>
> **Table 4: Comparison results of original Grounding DINO and our method under 10-shot setting.**
>
> | Method                  | ArTaxOr | Clipart1k |  DIOR  | DeepFish | NEU-DET |  UODD  | Avg|
> |:----------------------|:-------:|:---------:|:------:|:--------:|:-------:|:------:|:------:|
> | Grounding DINO(Swin-B)   |  72.0   |    58.4   |  38.2   |   35.2    |   26.5   |  28.6   |43.2|
> | Grounding DINO(Swin-B) + Ours   |  73.8   |    60.3    |  40.2   |   37.0   |   27.9   |  29.2   |44.7|

---

> > ### Comment · Reviewer_Xh4w · 2025-08-08
> >
> > I really appreciate the authors taking the time to do additional experiments and answer my questions in detail.
> >
> > The only remaining question I have is for the R3, did the authors test (learnable) weighted sum of features from different layers, instead of just simply linear addition? I'm curious how performant this simple baseline is since it's still in the vein of 'MoE' but in a non-conditional manner. Looking forward to seeing it included in the table as well.

---

> > > ### Author Response · Authors · 2025-08-08
> > > **Very happy that we addressed your concerns! Thanks for your insightful and valuable suggestions!**
> > >
> > > **Thank you for your positive feedback!**
> > >
> > > Below we address your concerns.
> > >
> > > >**Q1: Comparison of our method and simpler feature aggregation strategy.**
> > >
> > > **R1:** Thank you for your valuable suggestion.
> > >
> > > Following your suggestion, we add a simple feature aggregation strategy.
> > > + Learnable Weight: Introduce a learnable weight parameter to aggregate features from different layers of the VFM.
> > >
> > > This feature aggregation strategy introduces learnable weight parameters as a simple routing network to **select** and aggregate VFM features, thereby maintaining the MOE structure similar to ours. In contrast, the other three aggregation strategies (All Stages, Final Stage, Linearly Adding) simply aggregate the VFM features into the backbone features without selecting them. Therefore, it outperforms the other simple aggregation strategies, as shown in Table 1. However, it still leads to a decline in performance compared to our method, highlighting the importance of backbone features in guiding VFM feature selection. We will include these comparison experiments in the revised supplementary material.
> > >
> > > **Table 1: Comparison results of our method and simple feature aggregation strategies.**
> > >
> > > | Method                  | ArTaxOr | Clipart1k |  DIOR  | DeepFish | NEU-DET |  UODD  | Avg |
> > > |:----------------------|:-------:|:---------:|:------:|:--------:|:-------:|:------:|:------:|
> > > | All Stages    |  56.3  |   44.4      |  34.1   |   26.4    |   16.8   |  17.6   |32.6|
> > > | Final Stage      |   48.8    |  43.8   |  30.7   |  24.1    |   17.1   |  17.4   |30.3|
> > > | Linearly Adding    |  2.5   |   5.1    |  9.1   |  4.7   |   3.9   | 5.4   |5.1|
> > > | Learnable Weight  |  62.5   |   47.4    |   31.6  |  31.1   |  19.1   |  20.7 |35.4|
> > > | Ours   |  **71.3**   |   **49.9**    |  **37.8**   |  **34.1**    |  **23.7**   | **22.1**   |**39.8**|

---

### Official Review · Reviewer_cjXu · 2025-07-03

**Clarity:** 4
**Significance:** 4
**Originality:** 4
**Rating:** 5
**Confidence:** 4

**Summary:**

The authors presented a novel approach for Cross-Domain Few-Shot Object Detection (CD-FSOD) that effectively merges the localization abilities of the well-trained object detector with the strong generalized features learned by Vision Foundation Models (VFMs) to improve the detection performance. This is achieved by the Mixture-of-Experts (MoE) module that dynamically routes and modulates the integration of VFM features into the detector training. The MoE module includes two key components: 1)  a routing mechanism that assigns weights to VFM feature maps for each stage of the detector backbone, and 2) an expert projection (EP) module for aligning VFM features to those of detector features. The routing mechanism creates expert-wise weights and region-wise weights. The expert projection includes Shared and Private projections, which are concatenated to create VFM features aligned with backbone features. The aligned features are then processed by MoE with the help of routing weights. The method proposed in this paper can straightaway extend the capacity to detect novel classes without requiring re-training on the base data.

**Questions:**

1. In L90, the authors state, "DETR and its variants struggle on CD-FSOD tasks, primarily due to poor classification accuracy". Is there a reference to any papers or empirical results to justify this?
2. Please provide results on other backbones for the results in Table 1, especially with VIT-L/14, as most of the methods listed are using that backbone.
3. Why does the CLIP backbone give limited improvement compared to the DINO v2?

Minor
1. Repeated text about DINO DETR comparison in L239-243.
2. dAP value mismatch between Table 4 and the corresponding text in L248.

**Ethical Concerns:**

["NO or VERY MINOR ethics concerns only"]

**Final Justification:**

Overall, this paper was a strong submission from the beginning. My concerns were minor, and the authors have addressed all of them. So I maintain my rating to accept this paper.

**Limitations:**

The limitations are not discussed in the paper. However, the authors state that the paper has limitations. Please include them in the supplementary materials.

One obvious limitation is that when the number of layers of the detector and VFM increases, the routing computations increase quadratically.

**Paper Formatting Concerns:**

No formatting concerns

**Quality:**

4

**Strengths And Weaknesses:**

Strengths
1. The paper presents a conceptually simple solution for CD-FSOD by integrating the strong generalizable features of VFMs with the localization capacity of trained detectors to facilitate novel object detection with few-shot training.
2. Different from common FSOD methods, the technique proposed here doesn't need the detectors to be trained on the base classes. The authors used a pre-trained detector on the COCO dataset to adapt to a diverse set of new domains.
3. The paper is well written and easy to follow.
4. This method proposed is general purpose; any VFMs can be integrated into the standard detectors to perform FSOD using the proposed approach.


Weaknesses
1. It would be great to see results on other detection models, though it is not a significant weakness. A recent paper [1] with the GroundingDINO detector has good detection performance on the same problem settings, but not straightforward to compare with the detector and backbone used in this study.
2. It is not clear what is captured in the shared information and private information projection from the VFM features. The authors stated that the shared projection enables the aggregation of shared image information across different expert features. And the private projection module allows the model to learn tailored projection strategies for each expert feature. A good visualization is missing to understand this with more clarity.


[1] Enhance Then Search: An Augmentation-Search Strategy with Foundation Models for Cross-Domain Few-Shot Object Detection, CVPRw 2025.

---

> ### Author Rebuttal · Authors · 2025-07-31
>
> We sincerely thank Reviewer cjXu for the constructive feedback, insightful questions, and valuable suggestions!
>
> Below we address your concerns.
>
>  >**Q1: Comparison to ETS.**
>
> **R1**: Thank you for bringing ETS method [1] to our attention.
>
> We compare our method with ETS on six cross domain datasets. For fairness, we reproduce the experiments using the official ETS code and also report the results from the original paper. As shown in Table 1, our method can be directly applied to Grounding DINO without any modification and achieves similar performance to ETS, even though it is not specifically designed for Grounding DINO. In future work, we will explore designs better tailored to Grounding DINO. We will include these experimental results in the revised main paper.
>
> **Table 1: Comparison Results of ETS method and our method under 10-shot setting.**
>
> | Method                  | ArTaxOr | Clipart1k |  DIOR  | DeepFish | NEU-DET |  UODD  | Avg|
> |:----------------------|:-------:|:---------:|:------:|:--------:|:-------:|:------:|:------:|
> | ETS (Paper report)   |  71.2   |    61.5    |  37.5   |   44.1    |   26.1   |  29.8   |45.0|
> | ETS (Our implementation)   |  70.6   |    60.8    |  37.5   |   42.8    |   26.1   |  28.3   |44.4|
> | Grounding DINO (Swin-B) + Ours   |  73.8   |    60.3    |  40.2   |   37.0   |   27.9   |  29.2   |44.7|
>
> References:
>
> [1] Enhance Then Search: An Augmentation-Search Strategy with Foundation Models for Cross-Domain Few-Shot Object Detection, CVPRw 2025.
>
> >**Q2: Analysis of SEP and PEP modules.**
>
> **R2**: Thanks for your valuable question.
>
> The SEP module captures general saliency features that are common across all expert features. However, these feature maps lack the precise boundary details present in individual expert features. In contrast, our PEP module is tailored to extract those boundary cues unique to each expert, effectively filling in the missing boundary information. We validate our conclusion through PCA‑based feature visualizations. We will add visualization results in the revised main paper.
>
> >**Q3: Analysis of DETR’s classification capabilities.**
>
>
> **R3**: Thank you for your constructive question.
>
> We have demonstrated DETR’s relatively weak classification performance in Table 4 of main paper. We use the AP drop(dAP) caused by false‑positive samples to reflect the model’s classification capability. We show the experimental results in Table 2 below for easier checking. We identify notable weaknesses in the classification performance of DINO DETR.Specifically,the dAP values reach 44.45/40.12/35.68 under the 1/5/10-shot settings. This is consistent with our paper’s conclusion that DETR has relatively weak classification capability.
>
> **Table 2: Comparison results of classification capability.**
>
> | Method  | 1-shot FP loss ↓  | 5-shot FP loss ↓ | 10-shot FP loss ↓ |
> | :------------|:---------------:| :-----:|:-----:|
> | DINO DETR   | 44.45 | 40.12 | 35.68 |
> | DINOv2      | 29.73 |   27.70 | 30.26 |
> | Ours | 26.47       |   22.54 | 17.98 |
>
> >**Q4: Comparison results on different detector backbones.**
>
> **R4**: Thank you for your valuable suggestions.
>
> We conduct additional comparison experiments using differenet backbone architectures. A shown in Table 3, Applying our method to stronger backbones achieves better results than  result we report in the main paper. For example, integrating our method into DINO with a Swin‑B backbone improves the average performance by 2.4 AP, while using a ViT‑L backbone achieves a gain of over 4.6 AP. We will include these comparison experiments in the revised main paper.
>
> **Table 3: Comparison results of different backbone.**
>
> | Model                  | ArTaxOr | Clipart1k |  DIOR  | DeepFish | NEU-DET |  UODD  |Avg|
> |:----------------------|:-------:|:---------:|:------:|:--------:|:-------:|:------:|:------:|
> | DINO DETR(ResNet50)    |  71.3   |    49.9    |  37.8   |   34.1    |   23.7   |  22.1   |39.8|
> | DINO DETR(Swin-B)      |  75.4   |   56.7    |  39.5   |  35.1    |   23.2   |  23.1   |42.2|
> | DINO DETR(ViT-L)    |  75.8   |   60.3    |  42.0   |  37.2    |   25.1   | 25.9   |44.4|
>
> >**Q5: Analysis of different foundation models.**
>
> **R5:** Thank you for your valuable question.
>
> CLIP [4] uses large‑scale image text contrastive learning to capture high‑level semantics, excelling at zero‑shot classification and retrieval. However, it lacks the spatial granularity needed for region‑level tasks [1], making it less suitable for dense predictions such as detection and segmentation. In contrast, DINOv2 [5] uses self‑distillation with patch‑level reconstruction to learn fine‑grained, spatially coherent features. This leads to superior dense‑vision performance, as confirmed by studies [2,3] on visual grounding and segmentation.
>
> References:
>
> [1] Refining CLIP’s Spatial Awareness: A Visual‑Centric Perspective
>
> [2] From CLIP to DINO: Visual Encoders Shout in Multi‑modal LLMs
>
> [3] Learning Vision from Models Rivals Learning Vision from Data
>
> [4] Learning Transferable Visual Models From Natural Language Supervision (CLIP)
>
> [5] DINO v2: Learning Robust Visual Features Without Supervision
>
> >**Q6: Analysis of method limitations.**
>
> R6: Thank you for your valuable comment about our method limitation.
>
> The increased computational cost caused by the larger number of detector layers and expert features is indeed a limitation of our method. This limitation can be easily mitigated. As the number of backbone layers increases, we can sparsely select a subset of layers for processing, which has little impact on performance. Moreover, the routing mechanism introduces only minor computational cost and has minimal effect on inference speed. Another limitation of our approach is that it is currently designed only for visual tasks and does not yet handle multimodal scenarios, such as text‑guided visual grounding or video synthesis. However, our enhancement strategy can be easily applied to the visual backbones of multimodal models. As shown in Table 4, our method achieves strong performance when combined with Grounding DINO, even without explicit optimization for textual features. We will further explore these directions in future work.
>
>
> **Table 4: Comparison results of original Grounding DINO and our method under 10-shot setting.**
>
> | Method                  | ArTaxOr | Clipart1k |  DIOR  | DeepFish | NEU-DET |  UODD  | Avg|
> |:----------------------|:-------:|:---------:|:------:|:--------:|:-------:|:------:|:------:|
> | Grounding DINO(Swin-B)   |  72.0   |    58.4   |  38.2   |   35.2    |   26.5   |  28.6   |43.2|
> | Grounding DINO(Swin-B) + Ours   |  73.8   |    60.3    |  40.2   |   37.0   |   27.9   |  29.2   |44.7|

---

> > ### Comment · Reviewer_cjXu · 2025-08-03
> >
> > Thank you for your response. My concerns are adequately addressed. Very interesting results with Swin-B and ViT-L backbones.

---

> ### Author Response · Authors · 2025-08-03
> **Very happy that we addressed your concerns! Thanks for your insightful and valuable comments!**
>
> **Thank you for your positive feedback!**
>
> **We are delighted that we addressed your concerns.**
>
> **We also sincerely appreciate your efforts to improve our work.**

---

### Official Review · Reviewer_ZRE6 · 2025-07-07

**Clarity:** 3
**Significance:** 3
**Originality:** 3
**Rating:** 5
**Confidence:** 5

**Summary:**

This paper presents  a novel framework for cross domain few shot object detection by combining the detection models with vision foundation model in a mixture of experts (MOE) based method. Leveraging VFM and DETR through MOE framework enables improved generalization without loss of localization. In particular, it introduces a plug-in strategy that keeps a well-trained DETR detector frozen and boosts its generalization by injecting features from a vision foundation model, DINOv2. The proposed strategy introduces expert-wise & region-wise routers dynamically weight which DINOv2 layers and spatial regions matter for each DETR backbone stage as well as projection layers (shared and private) for feature alignment. Experimental study on different domains show improved performance over the counterpart methods

**Questions:**

I have listed all my concerns in the "Strength and Weaknesses section". Please refer to "Weaknesses" for the raised concerns and questions.

**Ethical Concerns:**

["NO or VERY MINOR ethics concerns only"]

**Final Justification:**

I would like to thank authors for effectively addressing my concerns. I have increased my score to 5.   Overall, I think this paper brings an important perspective, i.e. leveraging vision foundation models for enhanced downstream object detector performances without substantial computational sacrifice in inference time. I strongly encourage the authors to incorporate the arguments in the rebuttal, namely computational efficiency and comparisons with MLLMs and openvocab models.

**Quality:**

3

**Strengths And Weaknesses:**

Strengths

1)	The paper is well motivated and efficiently integrates vision foundation models with DETR architectures.

2)	The introduced novelties are in plug-and-play format, making them generalizable to various DETR methods

3) Experiments show consistent improvement over the baselines

Weaknesses

1)	Computational latency: How does the computational latency of the proposed method compared to counterpart methods

2)	Comparisons: How does the proposed method compared to open vocab methods(e.g. grounding-dino, yolo-world).

3)   How does  the proposed method perfroms compared to multi modal LLMs (MLLMs)  such as Qwen models or Ferret models which are well known with their visual grounding.

4)	No limitations are listed in the paper. What are the potential pitfall for proposed method.

---

> ### Author Rebuttal · Authors · 2025-07-31
>
> We sincerely thank Reviewer ZRE6 for the constructive feedback, insightful questions, and valuable suggestions!
>
> Below we address your concerns.
>
> >**Q1: Inference Speed Comparisons.**
>
> **R1**: Thanks for your insightful question.
>
> We compare inference speed with YOLO‑World and Qwen Model using Frames Per Second(FPS). Our method achieved similar FPS like YOLO‑World while attaining the highest performance, as shown in Table 1.
>
> **Table 1: Comparison results of inference speed and performance. 'mAP' denotes the mean performance across the six cross‑domain datasets.**
>
> |      Model       |  FPS  |  mAP  |
> |:----------------|:-----:|:-----:|
> |   Qwen Model     |  0.4  | 12.3  |
> |   YOLO‑World     |  5.8  | 13.5  |
> | DINO DETR + Ours |  5.6  | 39.8  |
>
> >**Q2&Q3: Comparison to MLLMs and OVMs.**
>
> **R2&R3**:  Thanks for your constructive suggestion.
> We compare our method with multimodal large language models(MLLMs) and open‑vocabulary methods(OVMs). Using their open‑source code, we conducted fair comparisons on the same dataset. Qwen model and Ferret model obtain results through text‑guided visual grounding. Grounding‑DINO and YOLO‑World derive detection results via image‑text matching. All codes are publicly available on their official websites.
>
> As shown in Table 2, our method achieved highest performance compared to OVMs and MLLMs across six cross domain datasets. Compared to Grounding‑DINO and YOLO‑World model, our method achieves the improvements of 22.1mAP/26.3mAP. For Qwen model and Ferret model, our method achieves the improvements of 27.5mAP/36.2mAP. We argue that models such as Qwen, despite being trained on large‑scale image‑text datasets, have never seen the novel classes in the cross domain dataset, resulting in weak zero‑shot performance. In contrast, our method adopts a cross‑domain learning strategy that integrates with visual foundation models, achieving superior performance on cross‑domain tasks. This also points to an important future direction, where we plan to extend our cross‑domain learning approach to MLLMs to further improve their performance on cross‑domain tasks.
>
> **Table 2: Comparison results of our method, MLLMs and OVMs under 10-shot setting.**
>
> | Model                  | ArTaxOr | Clipart1k |  DIOR  | DeepFish | NEU-DET |  UODD  | Avg|
> |:----------------------|:-------:|:---------:|:------:|:--------:|:-------:|:------:|:------:|
> | Qwen model             |  48.8   |    7.5    |  2.7   |   9.2    |   4.5   |  1.3   |12.3|
> | Ferret model          |     5.5    |    8.5       |    0.8    |     5.0     |    0.6     |    1.4    | 3.6 |
> | YOLO-World             |  10.5   |   37.5    |  3.1   |  29.5    |   0.1   |  0.2   |13.5|
> | Grounding DINO (Swin-B)     |  12.8   |   49.1    |  4.5   |  28.6    |   1.2   | 10.1   | 17.7|
> | DINO DETR (ResNet50) + Ours | 71.3 |   49.9    | 37.8   |  34.1    |  23.7   | 22.1   |39.8|
> >**Q4: Analysis of method limitations.**
>
> **R4**: Thanks for your insightful question.
>
> A key limitation of our method is that our current approach is limited to purely visual tasks and does not yet support multimodal scenarios such as text-guided visual grounding or video synthesis. Nonetheless, the proposed enhancement strategy is readily adaptable to the visual backbones of multimodal models. Even without optimizing textual features, our method still achieves strong performance when combined with Grounding DINO, as shown in Table 3. We plan to explore in future work.
>
>
> **Table 3: Comparison results of original Grounding DINO and our method under 10-shot setting.**
>
> | Method                  | ArTaxOr | Clipart1k |  DIOR  | DeepFish | NEU-DET |  UODD  | Avg|
> |:----------------------|:-------:|:---------:|:------:|:--------:|:-------:|:------:|:------:|
> | Grounding DINO (Swin-B)   |  72.0   |    58.4   |  38.2   |   35.2    |   26.5   |  28.6   |43.2|
> | Grounding DINO (Swin-B) + Ours   |  73.8   |    60.3    |  40.2   |   37.0   |   27.9   |  29.2   |44.7|

---

### Note · Authors · 2025-08-12

**We would like to sincerely thank all reviewers for your efforts and valuable comments to improve our work!**

**We have addressed all the questions raised by the reviewers and added relevant experiments to support our conclusions. We will include the experimental results and discussions in the revised main paper and supplementary materials.**

**We are pleased that all reviewers consistently appreciate our work's novelty, significance, method effectiveness, extensive experiments.**

**1. Reviewers appreciate our novelty and significance:**

+ The paper is well motivated and efficiently integrates vision foundation models with DETR architectures. (*Reviewer `ZRE6`*)
+ The introduced novelties are in plug-and-play format, making them generalizable to various DETR methods. (*Reviewer `ZRE6`*)
+ This method proposed is general purpose; Any VFMs can be integrated into the standard detectors to perform FSOD using the proposed approach. (*Reviewer `cjXu`*)
+ The motivation is clear: VFMs pretrained on large-scale data exhibit better generalization but lack accurate localization capabilities, while object detectors (OD) specialize in bounding box regression but are less robust to domain shifts. (*Reviewer `Xh4w`*)
+ The motivation is clear and meaningful. (*Reviewer `KWMR`*)


**2. Reviewers appreciate our method effectiveness and extensive experiments:**
+ Experiments show consistent improvement over the baselines. (*Reviewer `ZRE6`*)
+ The paper presents a conceptually simple solution for CD-FSOD. (*Reviewer `cjXu`*)
+ The design is straightforward yet effective, leveraging pretrained models without the need for retraining on base classes. (*Reviewer `Xh4w`*)
+ The method exhibits significant performance advantages over existing approaches on the CD-FSOD benchmark. It also showcases transferability across a variety of DETR architectures. (*Reviewer `Xh4w`*)
+ The proposed method is simple and effective, and it can be extended to various DETR-style detectors. (*Reviewer `KWMR`*)
+ The paper provides comprehensive analytical experiments. (*Reviewer `KWMR`*)

---

### Decision · Program_Chairs · 2025-09-17

**Decision:**

Accept (poster)

**Comment:**

Dear authors,

This draft has received all positive reviews. We strongly request authors to incorporate in the draft all the suggestions and required parts of discussion, that led to this decision.
For example,  (ZRE6) "computational efficiency and comparisons with MLLMs and openvocab models.", results appreciated by cjXu, (Xh4w) additional experiments that showed "the effectiveness of all modules introduced, further analyzed the experiments, and explained the rationale behind the design", (KWMR) comparison with Frozen DETR etc..
Xh4w further requested to improve the completeness.

Please provide clarify all the hyper-parameters, what they stand for, and what values were used in the experiments.
In the abstract its written"eliminates the traditional two-stage paradigm of base training", however, in introduction its stated that"without retraining on the source domain".  Clarity will improve readability.

We expect authors to provide essential and enough code that is necessary to reproduce their results, as we notice authors have promised "part of the code". Please note "The instructions should contain the exact command and environment needed to run to
reproduce the results. See the NeurIPS code and data submission guidelines (https://nips.cc/public/guides/CodeSubmissionPolicy) for more details.".


Congratulations

Regards

AC